# Simulation to Rules: A Dual-VLM Framework for Formal Visual Planning

**Yilun Hao**
MIT
`yilunhao@mit.edu`

**Yongchao Chen**
MIT / Harvard University
`yongchaochen@fas.harvard.edu`

**Chuchu Fan**
MIT
`chuchu@mit.edu`

**Yang Zhang**
MIT-IBM Watson AI Lab
`Yang.Zhang2@ibm.com`

## Abstract

Vision Language Models (VLMs) show strong potential for visual planning but struggle with precise spatial and long-horizon reasoning, while Planning Domain Definition Language (PDDL) planners excel at formal long-horizon planning but cannot interpret visual inputs. Recent works combine these complementary advantages by translating visual problems into PDDL. However, while VLMs can generate PDDL problem files satisfactorily, accurately generating PDDL domain files, which encode planning rules, remains challenging and typically requires human expertise or environment interaction. We propose VLMFP, a Dual-VLM-guided framework that autonomously generates both PDDL problem and domain files for formal visual planning. VLMFP combines a SimVLM that simulates action consequences with a GenVLM that generates and iteratively refines PDDL files by aligning symbolic execution with simulated outcomes, enabling multiple levels of generalization across unseen instances, visual appearances, and game rules. We evaluate VLMFP on 6 grid-world domains and demonstrate its generalization capability. On average, SimVLM achieves 87.3% and 86.0% scenario understanding and action simulation for seen and unseen appearances, respectively. With the guidance of SimVLM, VLMFP attains 70.0%, 54.1% planning success on unseen instances in seen and unseen appearances, respectively. We further demonstrate that VLMFP scales to complex long-horizon 3D planning tasks, including multi-robot collaboration and assembly scenarios with partial observability and diverse visual variations. Project page: https://sites.google.com/view/vlmfp.

## 1 Introduction

Although Large Language Models (LLMs) have shown strong performance in solving text-based planning problems (Wei et al., 2022; Yao et al., 2022; Raman et al., 2022; Yao et al., 2024), many real-world planning tasks, such as robot assembly, drone navigation, and autonomous driving, are inherently visual, making the reliance on carefully engineered text inputs impractical and limiting. This gap motivates the shift toward VLM-based planning, where visual inputs provide a more direct and intuitive basis for reasoning. However, current VLMs lack precise spatial understanding and long-horizon reasoning, which constrains their ability to address complex, multi-step planning problems that involve intricate spatial relationships among multiple objects (Wu et al., 2024).

On the other hand, Planning Domain Definition Language (PDDL) (McDermott, 2000) is a formal language designed to describe planning problems and domains in a structured, machine-interpretable way. PDDL has enabled numerous automated planners to derive long-horizon solutions. However, although PDDL-based planners excel at reasoning over structured domains, they depend on correctly structured PDDL domain and problem files and cannot directly interpret visual inputs. Constructing accurate PDDL definitions is non-trivial and requires expert knowledge, which is often inaccessible to non-expert users, limiting the broader adoption of PDDL planners in real-world scenarios.

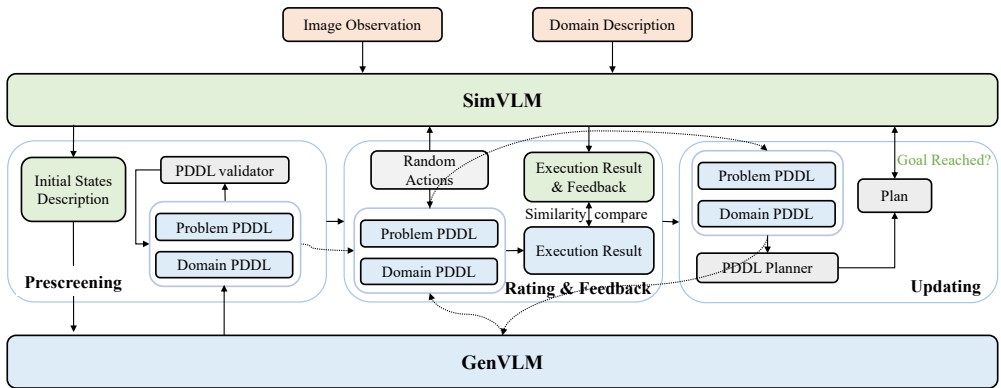

Figure 1: An overview of VLMFP. Sections in orange are the inputs. Sections in green represents SimVLM and its outputs. Sections in blue denotes the GenVLM and its outputs. Sections in gray are provided. VLMFP takes in the image and domain description, describe the scenario, generate and validate PDDL files, compare their execution results with SimVLM, and update the PDDL files.

Recent works have explored combining the advantages of language models and PDDL planners through various approaches. (Liu et al., 2023; Xie et al., 2023) employ LLMs as translators to convert natural language scenario descriptions into PDDL problem files. (Mahdavi et al., 2024) leverages environment interactions to enable file generation for both problem and domain. However, these methods require either textual scenario descriptions, environment access, or pre-defined PDDL files, which cannot be directly achieved through visual inputs. More recently, vision-language models (VLMs) have been used to extract scenario information from visual inputs and generate corresponding PDDL problem files(Shirai et al., 2024; Dang et al., 2025). However, since current VLMs lack the ability to accurately generate the domain PDDL, these approaches also assume access to ground truth domain PDDL files, without which PDDL planners cannot produce any results.

In this paper, we address the challenge of visual long-horizon planning by introducing a novel framework, **VLM**-Guided **F**ormal **P**lanning (VLMFP, illustrated in Fig. 1), a Dual-VLM-guided framework that autonomously generates both problem and domain PDDL files for visual planning. VLMFP integrates two specialized VLMs: a fine-tuned SimVLM that perceives the scenario from visual inputs and simulates action outcomes, and a large GenVLM that generates and iteratively refines PDDL files by aligning their execution with SimVLM's simulations. Generating both problem and domain PDDL from visual inputs requires object recognition, spatial understanding, reasoning, and PDDL knowledge. We fine-tune a small VLM as SimVLM to strengthen its spatial reasoning, while using a large model as GenVLM for general reasoning and extensive PDDL knowledge. This formulation is agnostic to environment dimensionality, allowing the same perception-to-symbolic pipeline to extend beyond grid-based settings to visually grounded 3D planning tasks.

Importantly, VLMFP achieves multiple levels of generalizability. A generated domain PDDL can be reused across all instances of the same domain, while problem PDDL files can be adapted efficiently for new instances as in-context examples. The framework also transfers well to unseen appearances and even altered environment rules. We evaluate VLMFP on six grid world domains, showing that SimVLM reliably describes scenarios, simulates actions, and determines goal achievement, while GenVLM, guided by SimVLM feedback, generates valid PDDL files that enable planners to solve both seen and unseen problems. Beyond grid-world benchmarks, we further examine the scalability of VLMFP on complex long-horizon 3D planning tasks, including assembly and multi-robot collaboration scenarios under partial observability.

In summary, our key contributions are:

- We construct a large-scale dataset of 430k action sequence simulations with reasoning and feedback across 6 grid-world domains of different map sizes, appearances, and game rules. We fine-tune Qwen2-VL-7B with the dataset, and our finetuned model demonstrates strong generalization to unseen instances, appearances, and game rules.
- We propose VLMFP, a Dual-VLM-guided framework that autonomously generates PDDL domain and problem files for visual planning, which, to our knowledge, is the first framework to leverage visual inputs to generate both PDDL files without human feedback or direct environment access.

- By combining SimVLM (for perception and action simulation) with GenVLM (for symbolic reasoning and file refinement), VLMFP achieves robust, reusable domain generation and efficient problem instantiation. VLMFP notably achieves 70.0% and 54.1% success rates with GPT-4o as the GenVLM, outperforming the best baseline CodePDDL$_{\text{GPT-4o}}$ by 39.3% and 21.8% for unseen instances in seen and unseen appearances, respectively for 6 grid-world domains.
- We demonstrate that VLMFP scales beyond grid-world benchmarks to complex long-horizon 3D tasks with partial observability and diverse visual variations, with an average of 86.4% and 79.8% success rates for unseen instances in seen and unseen appearances, respectively.

## 2 RELATED WORKS

**LLM and VLM Planning** Although LLMs have achieved impressive performance on text-based planning tasks (Wei et al., 2022; Yao et al., 2022; Raman et al., 2022; Yao et al., 2024), many real-world planning problems are inherently visual. Relying solely on carefully constructed text inputs is often impractical and cannot capture rich spatial details in visual environments. This limitation has motivated a growing interest in VLM-based planning, where visual observations provide a more direct and intuitive foundation for reasoning (Driess et al., 2023; Huang et al., 2023; Zhang et al., 2023; Nasiriany et al., 2024). However, current VLMs still fall short in precise spatial understanding (Wu et al., 2024), restricting their effectiveness on complex planning tasks involving multiple objects and intricate spatial relationships. Recent works explore Visual Chain-of-Thought (Li et al., 2025; Zhao et al., 2025), using images as part of the reasoning process while generating plans. However, this process introduces computational overhead and is hard to apply to long-horizon planning tasks. In this work, we fine-tune SimVLM to enhance its visual–spatial understanding and reasoning, enabling it to guide the generation of PDDL files for solving long-horizon planning problems.

**LLM and VLM + PDDL** Since existing LLMs lack the ability to reliably perform long-horizon reasoning in complex tasks (Achiam et al., 2023a; Valmeekam et al., 2022; 2023; Kambhampati et al., 2024), recent works have explored combining LLMs with external solvers to augment their reasoning and planning capabilities (Wu et al., 2022; He-Yueya et al., 2023; Pan et al., 2023; Ye et al., 2024; Li et al., 2023; 2024; Hao et al., 2024a;b). Among such solvers, combining LLMs with PDDL-based planners is a powerful option (Silver et al., 2022; Liu et al., 2023; Stein & Koller, 2023; Xie et al., 2023; Stein & Koller, 2023; Guan et al., 2023; Oswald et al., 2024; Mahdavi et al., 2024), as PDDL is designed for precise symbolic reasoning over long-horizon problems. However, these works either require predefined PDDL domain files, human corrections, or environment access to provide feedback while generating PDDL files. In addition, to further enable visual interpretation, many works combine VLMs with PDDL planners (Shirai et al., 2024; Dang et al., 2025) by generating PDDL Problem files based on image observation. However, similarly, as generating PDDL domain files is complicated, these works assume access to predefined domain files. In our work, we use a Dual-VLM-guided framework to autonomously generate both PDDL problem and domain files, without human guidance or environment access.

## 3 VLMFP

### 3.1 PROBLEM FORMULATION AND CHALLENGES

In our setting, a visual planning problem is defined by two pieces of information: ❶ A **domain description** $n_d$, which is a natural language description of the general rules, domain settings, available actions, and the planning objective of the problem; and ❷ a **problem setup image** $i_p$, which is an image showing the layouts, initial states of the problem instance.

Figure 2 top left shows an example domain description and problem setup image of the planning problem *FrozenLake*. As can be observed, the domain description elaborates the rules of the game (*e.g.*, no stepping onto an ice hole) and available actions of the player (*i.e.*, moving up, down, left, and right). The problem setup image is a map showing the initial positions of the player, the destination, and frozen likes.

The goal of VLMFP is, given the problem definition pair $(n_d, i_p)$, to come up with an action sequence to achieve the goal. Considering the complementary strengths of VLMs in image perception and natural-language interpretation, and symbolic methods such as PDDL in formal planning, our

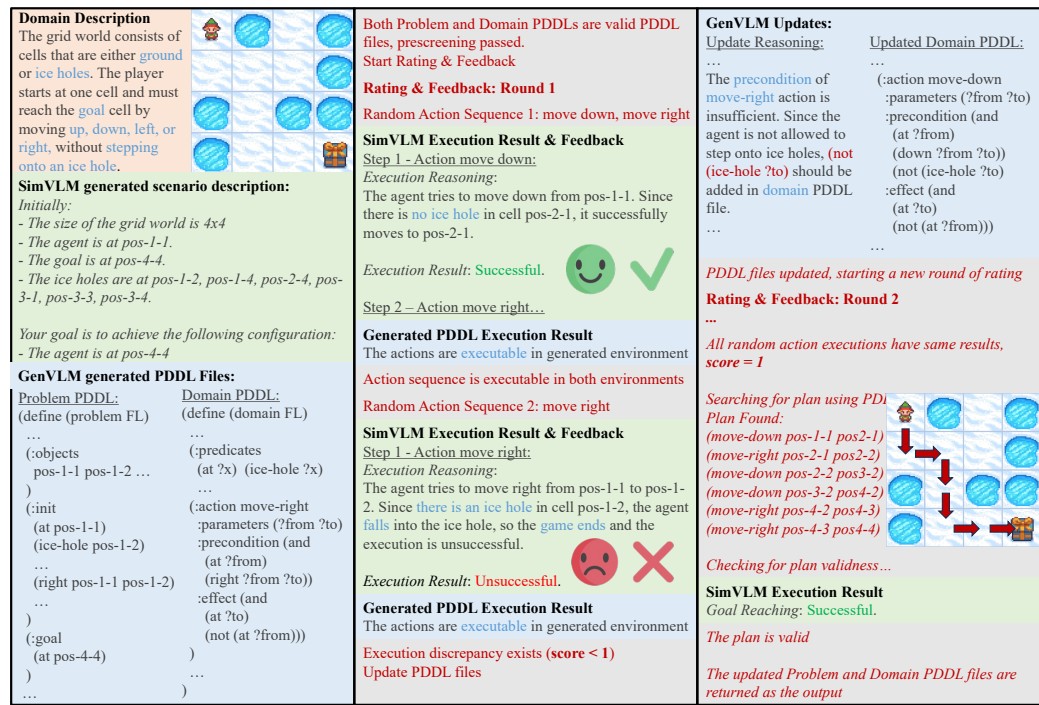

Figure 2: An example of how VLMFP tackle a Frozenlake instance.

framework integrates these advantages to transform visual observations into formal specifications that can be solved symbolically.

However, translating visual inputs into PDDL specifications using VLMs is an error-prone process. PDDL provides a standardized framework for representing automated planning problems, consisting of two files: ❶ a **PDDL domain file** $f_d$, which formally encodes rules by defining predicates and actions with preconditions and effects; and ❷ a **PDDL problem file** $f_p$, which specifies objects, the initial state, and the goal for a concrete instance. $f_d$ and $f_p$ can be viewed as formal translations of the domain description $n_d$ and problem setup image $i_p$, respectively. See Appendix E for examples.

The translations of these two files using VLMs come with their respective challenges. On one hand, translating the problem setup from the image $i_p$ to the PDDL problem file $f_p$ requires a precise understanding of the spatial relationship in the image, but existing VLMs tend to make mistakes in spatial perception tasks such as counting rows/columns, measuring distance *etc* (Wu et al., 2024). On the other hand, the PDDL domain file $f_d$ should be general enough to work for all problem instances under the same task (*e.g.*, different maps of the same FrozenLake game), but generating the PDDL domain file using VLM alone, without human input or constant access to the environment, can easily fail to accurately model all constraints and dynamics(*e.g.*, to move left, the goal cell must not be ice hole, and must be at left of the origin cell).

## 3.2 VLMFP OVERVIEW

To tackle the aforementioned challenges, VLMFP introduces a dual-VLM framework, consisting of a **SimVLM** (Simulation VLM) and a **GenVLM** (Generation VLM). SimVLM takes the domain description $n_d$, the problem setup image $i_p$, and a list of action $\pi = [a_{1:T}]$ (*e.g.*, move left in the FrozenLake game), and performs three tasks: ❶ Describing the spatial relationship in the $i_p$ in natural language, denoted as $n_p$, ❷ reasoning and predicting the consequence of the proposal actions (*e.g.* whether the player hits a wall/ice hole), and ❸ determining whether executing $\pi$ successfully achieves the problem goal. GenVLM generates both PDDL files with the help of SimVLM.

The two VLMs should have different comparative advantages. SimVLM should be stronger at precisely understanding the spatial relationship in images, and have a superior capability in simulating

action consequences. GenVLM should possess stronger general reasoning and question-answering capabilities, and have richer knowledge in PDDL.

With these complementary strengths, SimVLM supports GenVLM in two key ways. First, it provides a structured description of the spatial relationships in the problem setup image $i_p$, compensating for GenVLM's weaker spatial perception. Second, SimVLM serves as an execution-checking oracle: by comparing its simulated action outcomes with those derived from the generated PDDL model, any discrepancies signal potential errors and are fed back to GenVLM for refinement.

Based on these rationales, VLMFP generates PDDL files via the following four steps (as in Figure 1):

**Step 1: Candidate PDDL files generation.** Given the domain descriptions and problem setup image, $(n_d, i_p)$, the SimVLM first generates a description of the spatial relationships $(n_p)$ in the problem setup image, which is then fed to the GenVLM to generate candidate PDDL files, namely

$$n_p = V_S(n_d, i_p), \quad f_d^{(0)}, f_p^{(0)} = V_G(n_d, i_p, n_p), \tag{1}$$

where $V_S$ and $V_G$ represent the forward passes of SimVLM and GenVLM, respectively.

**Step 2: Prescreening.** The generated PDDL files are checked against syntactic correctness and semantic consistency.

**Step 3: Simulation consistency checking.** Random action sequences are executed in the PDDL environment based on the generated PDDL files, whose results are compared against the simulation results by SimVLM. Inconsistencies are summarized as feedback, denoted as $s$.

**Step 4: PDDL files updating.** GenVLM refines the generated PDDL files based on the feedback:

$$f_d^{(t)}, f_p^{(t)} = V_G\big(n_d, i_p, n_p; s, f_d^{(t-1)}, f_p^{(t-1)}\big). \tag{2}$$

VLMFP iterates over steps 2-4 until consistency is achieved and a plan is found by the PDDL planner. Figure 2 gives an example of this process. In the following, Section 4.1 will discuss the implementation details of the two VLMs; Section 4.2 will provide further details of the above steps.

# 4 IMPLEMENTATION DETAILS

## 4.1 VLM DETAILS

**SimVLM.** As discussed in Section 3.2, SimVLM is tasked with precisely describing the problem setup image $i_p$ and predicting action consequences. Since existing VLMs are generally weak in spatial relationship reasoning, we fine-tune Qwen2-VL-7B to accomplish these tasks. Specifically, we collected six grid world domains of varying complexity. For each domain, we collect data across map sizes ranging from 3 to 8 with varying obstacle probabilities, and create 5–6 distinct visual appearances. In total, this yields a dataset of 430k datapoints.

The fine-tuning process enables SimVLM to better align visual observations with spatially grounded reasoning. In particular, SimVLM learns to generate concise natural language narratives of the initial scenario and to simulate the outcomes of action sequences within these domains. By grounding visual inputs in structured textual descriptions and action simulations, SimVLM serves as a reliable intermediate model that bridges raw perception and the formal representation required for PDDL generation. We show examples of API-based large VLMs' failure to accurately describe the scenario and simulate actions in Appendix B.4.

**GenVLM.** While SimVLM is tailored for perception and structured simulation, generating PDDL domain and problem files requires advanced reasoning to capture action dynamics and a precise understanding of PDDL syntax and conventions to ensure valid domain specifications. To address this, we leverage a large VLM API, GPT-4o (Achiam et al., 2023b), referred to as GenVLM, which has broader reasoning capacity and linguistic knowledge needed for reliable PDDL construction. Guided by the scenario descriptions and simulated outcomes provided by SimVLM, GenVLM generates initial PDDL files and iteratively refines them to resolve inconsistencies. In this way, GenVLM complements SimVLM by bridging high-level reasoning with formal symbolic representation.

## 4.2 Algorithmic Details

**Prescreening.** In this step, VLMFP generates and verifies whether the generated initial PDDL problem and domain files are structurally valid before moving on to further evaluation. A PDDL domain or problem is valid if it is syntactically correct and semantically consistent, meaning the domain's actions, predicates, and types are well-defined and the problem's objects, initial state, and goals align with that domain.

For example, as shown in Fig. 2, in the FrozenLake domain, the problem PDDL defines ice hole positions using predicate `ice-hole`, but if the Domain PDDL does not define `ice-hole` under the `(:predicates` section, these two files would be identified as invalid and cannot pass the prescreening. We set the maximum rounds of file regeneration to be 5. This stage is crucial because it filters out syntactic or structural errors at an early step, thereby ensuring that only valid PDDL files progress to the consistency checking process and that subsequent comparisons are meaningful.

**Simulation consistency checking.** In this step, VLMFP evaluates the fidelity of the generated PDDL files by comparing their execution results with those of SimVLM. Random executable action sequences are sampled in both environments and executed in the other environment: SimVLM simulates the outcomes and provides reasoning step by step, and the PDDL environment checks action executability under the generated domain.

For a walk length $T$, let $P_{\text{sim},T}$ be a distribution over SimVLM-executable $T$-step action sequences, where $E_{f_d,f_p}(q) \in \{0,1\}$ indicates whether $q$ is executable in the generated PDDL environment $(f_d, f_p)$. Similarly, let $P_{f_d,f_p,T}$ be a distribution over $T$-step walks executable in $(f_d, f_p)$, and let $E_{\text{sim}}(q) \in \{0,1\}$ indicate SimVLM executability. Following (Mahdavi et al., 2024), we define the Exploration Walk (EW) score as:

$$
m_{\text{EW}}(\hat{d}, \hat{p}) = 2 \bigg( \Big( \frac{1}{T_{\max}} \sum_{T=1}^{T_{\max}} \mathbb{E}_{q \sim P_{\text{sim},T}} [E_{f_d,f_p}(q)] \Big)^{-1} + \Big( \frac{1}{T_{\max}} \sum_{T=1}^{T_{\max}} \mathbb{E}_{q \sim P_{f_d,f_p,T}} [E_{\text{sim}}(q)] \Big)^{-1} \bigg)^{-1}
$$
(3)

The EW score compares bi-directional similarity. A higher EW score indicates stronger alignment between the generated and reference environments. For example, in the FrozenLake domain, SimVLM predicts that moving right from pos-1-1 to pos-1-2 should fail due to an ice hole, whereas the generated PDDL environment instead allows the move. This discrepancy lowers the EW score and signals GenVLM to refine the domain file. Importantly, this stage not only enables VLMFP to detect execution mismatches and rate generated files using the EW score, but also produces natural language feedback on the incorrect actions to guide further file updates.

**PDDL files updating.** In this step, VLMFP refines the generated PDDL files based on the discrepancies identified in the last step. Using the feedback that describes the incorrect action and its expected outcome, GenVLM systematically inspects all critical components by listing and reasoning whether the objects, object types, init states, goal states, predicates, actions are valid and coherent, identifies which file(s) are erroneous, and applies targeted modifications, such as adding missing objects or correcting action preconditions, to regenerate updated PDDL specifications.

## 5 Experimental Results

### 5.1 Domains

We finetune SimVLM and test on six grid world domains: **Frozenlake**, **Maze**, **Sokoban**, **Package**, **Printer**, and **Overcooked** (Towers et al., 2024; Silver & Chitnis, 2020; Jin et al., 2023; Wu et al., 2021). For each domain, we collect data across map sizes ranging from 3 to 8 with varying obstacle probabilities, and create 5–6 distinct visual appearances. In total, this yields a dataset of 430k datapoints. By default, we use this dataset for SimVLM fine-tuning, which is used to evaluate framework VLMFP. We also evaluate on two more complex 3D embodied tasks, **MultiRob** and **Assembly** (Feng et al., 2025; Ji et al., 2025), to assess generalization across environment complexity and visual variability. Domain descriptions, visualizations of different scenario appearances, domain complexity analysis, and dataset statistics are included in Appendix A.1, A.2, A.3 and B.5.

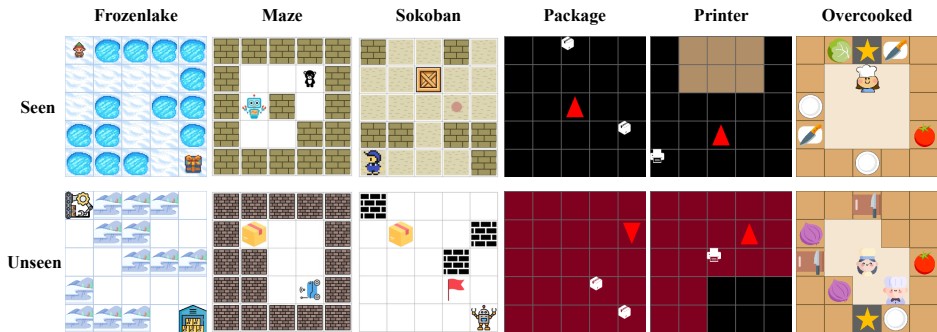

Figure 3: Visualizations of some seen and an unseen appearances for six grid world domains.

## 5.2 SimVLM Performance and Appearance Generalization

We fine-tune a Qwen2-VL-7B (Wang et al., 2024) model as SimVLM on datasets from six grid world domains. Given a domain description, an image, and an action sequence, SimVLM outputs (1) a natural language problem description, (2) step-by-step reasoning and outcomes, and (3) a judgment on goal achievement. Appendix B provides fine-tuning details, sample inputs and outputs. Before integrating SimVLM into VLMFP, we evaluate its performance on these tasks individually. Since each datapoint has a fixed-format output, we measure performance by exact string matching between the model's predictions and the ground-truth outputs. The four outputs are referred to as **Task Description**, **Execution Reason**, **Execution Result**, and **Goal Reach**.

To assess SimVLM's generalization ability, we consider two settings: Seen (S) and Unseen (U) appearance. S evaluates images with visual styles present in training, while U evaluates novel appearances under the same rules. This measures how well SimVLM fits the training distribution and how robustly it transfers to new visual variations of the same task. For each domain, we evaluate on 1000 random datapoints for both S and U, and report the average string matching rate.

Table 1: **String matching rate** (%) for 4 SimVLM output types on 6 grid world domains.

| Output | Frozenlake | | Maze | | Sokoban | | Package | | Printer | | Overcooked | | Average | |
|---|---|---|---|---|---|---|---|---|---|---|---|---|---|---|
| | S | U | S | U | S | U | S | U | S | U | S | U | S | U |
| **Task Descrip.** | 100 | 92.3 | 99.1 | 89.8 | 74.9 | 51.6 | 100 | 99.7 | 99.9 | 96.3 | 99.2 | 65.6 | 95.5 | 82.6 |
| **Exec. Reason** | 96.3 | 96.0 | 85.3 | 97.1 | 76.6 | 84.6 | 89.8 | 88.4 | 87.4 | 86.8 | 78.9 | 75.9 | 85.7 | 88.1 |
| **Exec. Result** | 96.2 | 95.9 | 85.3 | 96.8 | 75.2 | 83.4 | 89.8 | 88.4 | 87.4 | 86.8 | 78.8 | 75.7 | 85.5 | 87.8 |
| **Goal Reach** | 94.0 | 93.7 | 80.3 | 94.7 | 74.2 | 83.2 | 87.8 | 86.0 | 84.1 | 84.6 | 73.9 | 71.2 | 82.4 | 85.6 |
| Average | 96.6 | 94.5 | 87.5 | 94.6 | 75.2 | 75.7 | 91.9 | 90.6 | 89.7 | 88.6 | 82.7 | 72.1 | 87.3 | 86.0 |

**Results and Analysis.** We summarize SimVLM's performance on six grid-world domains under both seen (S) and unseen (U) appearance settings in Table 1. There are three key takeaways:

First, SimVLM achieves strong accuracy across tasks on seen visual appearances. On average, it reaches 95.5%, 85.7%, 85.5%, 82.4% in the seen setting across all domains for the four outputs, respectively. This highlights the effectiveness and reliability of SimVLM.

Second, SimVLM generalizes well to unseen visual appearances. On average, it reaches 82.6%, 88.1%, 87.8%, 85.6% in the unseen setting across all domains for the four outputs, respectively. The gap between S and U is minimal (average drop of 1.3% across all domains and outputs), showing that the fine-tuned model is not overfitting to training appearances and can transfer to new visual styles while maintaining stable output quality. This robustness is crucial for scaling to real-world environments where visual variation is common.

Third, errors in the Task Description do not necessarily affect action simulation. While the Task Description output under U drops by 12.9% on average compared to S, the execution simulation outputs remain equally accurate. This suggests that failures in Task Description typically involve only a small subset of objects, which often do not impact the sampled action sequences. For instance, the Sokoban task contains multiple object types: box, goal, wall, agent, floor, and the appearances

of these objects are all changed. When the appearances of the objects in a domain change, since SimVLM is not familiar with the new look of objects, it could fail to recognize or localize some objects. However, when the misidentified objects are not directly involved in the sampled action sequence, execution reasoning and results remain correct. This separation highlights that SimVLM is robust in reasoning about action dynamics, even when its initial object recognition is imperfect.

Together, these results validate that SimVLM can produce reliable structured outputs from visual inputs and maintain strong generalization to new appearances, making it a suitable foundation for guiding PDDL generation in VLMFP.

## 5.3 VLMFP PERFORMANCE

We evaluate VLMFP on six domains using GPT-4o (Achiam et al., 2023b). Each problem instance comes with a domain description and an image observation, from which VLMFP generates both PDDL problem and domain files. The problem file is then used as an in-context example to prompt GenVLM to generate PDDL problem files for 100 new instances, with the PDDL Planner validating the plans. We repeat this process for 15 different input instances and calculate the average success rate of finding correct plans, that is, the average success rate of 1500 trials. We follow the same procedure for unseen appearances. Appendix C provides all prompts and failure-case analysis.

**Baselines** We compare VLMFP against 1) Direct: VLM direct plan generation with an in-context example, 2) CoT: chain-of-thought prompting (Wei et al., 2022) with an in-context example by asking LLMs to reason before generating the final answer, 3) CodePDDL: prompts LLM to generate PDDL files, given problem descriptions generated by SimVLM. For all baselines, we use GPT-4o and also include Direct and CoT baselines with GPT-5 (OpenAI, 2025). All baselines have the same domain description and instance image observation as inputs. For Direct and CoT, since they do not generate PDDL files, we directly apply them to generate plans for the 100 problem instances.

Table 2: **Success rate** (%) comparison of VLMFP with baselines on 6 grid world domains.

| Method | Frozenlake | | Maze | | Sokoban | | Package | | Printer | | Overcooked | | Average | |
|---|---|---|---|---|---|---|---|---|---|---|---|---|---|---|
| | S | U | S | U | S | U | S | U | S | U | S | U | S | U |
| Direct$_{\text{GPT-4o}}$ | 7.0 | 8.0 | 1.0 | 2.0 | 0.0 | 0.0 | 0.0 | 0.0 | 0.0 | 0.0 | 0.0 | 0.0 | 1.3 | 1.7 |
| Direct$_{\text{GPT-5}}$ | 30.0 | 24.0 | 10.0 | 15.0 | 15.0 | 9.0 | 5.0 | 3.0 | 11.0 | 9.0 | 0.0 | 0.0 | 11.8 | 10.0 |
| CoT$_{\text{GPT-4o}}$ | 9.0 | 10.0 | 1.0 | 2.0 | 0.0 | 0.0 | 0.0 | 0.0 | 0.0 | 0.0 | 0.0 | 0.0 | 1.7 | 2.0 |
| CoT$_{\text{GPT-5}}$ | 36.0 | 28.0 | 14.0 | 16.0 | 14.0 | 13.0 | 2.0 | 3.0 | 10.0 | 12.0 | 0.0 | 0.0 | 12.7 | 12.0 |
| CodePDDL$_{\text{GPT-4o}}$ | 88.1 | 77.1 | 87.5 | 74.9 | 0.0 | 0.4 | 0.0 | 19.0 | 7.2 | 22.1 | 1.1 | 0.0 | 30.7 | 32.3 |
| **VLMFP $_{\text{GPT-4o}}$** | **95.2** | **81.1** | **88.7** | **82.8** | **55.8** | **25.1** | **75.2** | **53.4** | **58.7** | **61.0** | **46.2** | **21.3** | **70.0** | **54.1** |

**Results and Analysis** We present VLMFP performance in Table 2. There are three key takeaways:

First, VLMFP achieves strong performance, significantly outperforms all baselines. Across six domains, VLMFP achieves an average success rate of 70.0% (S) and 54.1% (U), while the strongest baseline (CodePDDL$_{\text{GPT-4o}}$) reaches 30.7% (S) and 32.3% (U). Note that CodePDDL$_{\text{GPT-4o}}$ has access to the SimVLM generated scenario descriptions, which substantially aid the problem generation. Direct plan generation and chain-of-thought prompting perform very poorly, confirming that without explicit PDDL grounding, LLMs struggle to maintain consistency in long-horizon planning. These results highlight the advantage of plan generation with formal PDDL files guided by SimVLM. Importantly, the results demonstrate that, after seeing one problem instance in the domain, VLMFP can reliably aid the planning of all problem instances not seen by VLMFP. This clearly proves VLMFP to be an effective, generalizable, and efficient framework.

Second, VLMFP generalizes well to unseen appearances. On average, VLMFP can solve 54.1% of the unseen problem instances in unseen appearances, which still substantially outperforms the baselines. This demonstrates that the strong perception capability of SimVLM is well adapted by VLMFP, enabling VLMFP to generalize better across different visual styles.

Third, domain complexity influences the relative gains of VLMFP. In simpler domains such as Frozenlake and Maze, both VLMFP and CodePDDL achieve high success rates, though VLMFP still maintains a clear margin. In complex domains like Sokoban and Printer where reasoning about different object types and multiple complicated actions is required, VLMFP delivers a strong perfor-

mance gain over CodePDDL: 55.8% vs. 0.0% (U) in Sokoban, and 58.7% vs. 7.2% (U) in Printer. This shows that the rating and updating processes are crucial for capturing complex domains.

Together, these findings validate that VLMFP is not only effective and efficient in solving unseen problem instances, but also robust to variations in visual appearances and domain complexity.

## 5.4    VLMFP EFFECTIVENESS OF VLMFP COMPONENTS

We validate each stage of LLMFP with ablations on 6 domains. We examine the effectiveness of Prescreening, Feedback, and Update by removing them from VLMFP one at a time. Similarly, VLMFP generate PDDL files based on one problem instance and test with 100 other problem instances.

Table 3: **Success rate** (%) when removing key components of VLMFP on 6 grid world domains.

| Method | Frozenlake | Maze | Sokoban | Package | Printer | Overcooked | Average |
|---|---|---|---|---|---|---|---|
| **No Prescreening** | 94.3 | 62.5 | 23.9 | 56.9 | 37.3 | 10.3 | 47.5 |
| **No Feedback** | **95.5** | 84.9 | 38.0 | 58.9 | 53.4 | 35.9 | 61.1 |
| **No Update** | 88.1 | 87.5 | 0.0 | 0.0 | 7.2 | 1.1 | 30.7 |
| **VLMFP** | 95.2 | **88.7** | **55.8** | **75.2** | **58.7** | **46.2** | **70.0** |

We report ablation results in Table 3. Removing any of the three components negatively impacts performance across domains. While prescreening and feedback contribute to steady improvements, the updating stage is essential. Success rates reduce to near zero in complex domains without updating. These demonstrate the effectiveness of all three stages for VLMFP to achieve robust performance.

## 5.5    GENERALIZATION ON GAME RULES

Beyond generalization to unseen instances and visual appearances, we further evaluate whether the framework can adapt to changes in underlying environment rules. In FrozenLake, we construct 15 rule variants by modifying core mechanics such as ice-hole behavior and movement dynamics (see Appendix D for rule descriptions). A SimVLM model is fine-tuned with these variations and evaluated on five unseen rules using 100 problem instances. The unseen rules introduce progressively more complex dynamics, ranging from reworded known behaviors and modified transition effects to novel mechanics that influence multi-step execution (Appendix D). We evaluate whether SimVLM can correctly predict execution reasoning and outcomes for actions involving these unseen rules.

Table 4: **Success rate** (%) when testing SimVLM on unseen rules for Frozenlake.

| Output | Rule1 | Rule2 | Rule3 | Rule4 | Rule5 |
|---|---|---|---|---|---|
| **Exec. Reason & Result** | 94.2 | 99.0 | 76.1 | 59.2 | 71.1 / 0 |

Results are summarized in Table 4. SimVLM achieves strong performance on Rules 1–4, reaching accuracies of 94.2%, 99.0%, 76.1%, and 59.2%, respectively, demonstrating robust generalization to altered rule formulations and modified environment dynamics. These results indicate that SimVLM can successfully interpret and apply previously unseen rule variations when their underlying structure remains related to training dynamics. Performance drops on Rule 5, which introduces a completely novel freezing mechanic not observed during training. Although SimVLM correctly explains the rule behavior (i.e., the next action should be skipped) with 71.1% success rate, it fails to apply this constraint during execution, resulting in correct reasoning but unsuccessful action simulation. This highlights a key limitation when adapting to entirely new dynamics, while also suggesting that SimVLM retains partial semantic understanding of unseen rules.

Overall, these results demonstrate that SimVLM exhibits promising generalization to modified and unseen environment rules, supporting the adaptability of VLMFP beyond fixed planning dynamics.

## 5.6    VLMFP FOR COMPLEX 3D PLANNING TASKS

While previous experiments focus on controlled grid-world domains for systematic evaluation, VLMFP is not restricted to such settings and naturally extends to complex 3D continuous environments. We further evaluate our framework on two complex long-horizon 3D tasks: an assembly

task (Feng et al., 2025) and a multi-robot collaboration task (Ji et al., 2025), shown in Figure 4. The domains are challenging because (1) the tasks are fully 3D with no natural-language goal, requiring SimVLM to infer spatial relationships and relational dependencies directly from images, and (2) all block attributes (number, color, shape, position, state), table textures and colors, robot and background colors, and dependency orders are randomized with frequent occlusions, producing highly variable, partially observable configurations. We collect 100k datapoints for each domain and fine-

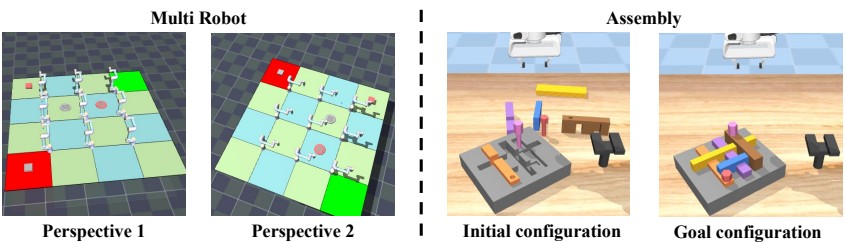

Figure 4: Visualization of MultiRob and Assembly.

tune Qwen2-VL-7B as our SimVLM(example inputs and outputs are provided in Appendix B.7). Appearance variation is introduced by randomizing robot, table, background, and object colors, lighting conditions, and camera viewpoints in MultiRob, and by varying block color, position, order, and number under different random seeds in Assembly. Unseen appearances use colors sampled from a different distribution together with unseen random seeds. Following our evaluation protocol, we evaluate SimVLM on 1,000 datapoints each from seen (S) and unseen (U) appearance distributions, and report the end-to-end success rates of VLMFP over 1,500 trials for each setting.

**Results and Analysis** As shown in Table 5, SimVLM achieves consistently strong performance across all four evaluation metrics, with an average of 96.2%, 96.0% for Assembly and 99.5%, 99.5% for MultiRob for seen and unseen appearances, respectively. This strong SimVLM performance greatly supports PDDL files generation, with VLMFP 's end-to-end performance: the overall planning success rates are high across both seen and unseen settings for both tasks, achieving 92.4%, 85.9% for Assembly and 80.3%, 73.6% for MultiRob, respectively.

Since SimVLM can already reason over initial and goal images to infer object dependencies, as well as multi-view observations to address occlusions, the framework naturally supports further extensions. For example, incorporating temporal image sequences could enable reasoning over scene dynamics and support planning in evolving environments. Extending our framework to more diverse and realistic 3D tasks is an important direction for future work, particularly given the growing availability of large-scale real-world vision and robotics datasets.

Table 5: SimVLM **String matching rate** (%) and VLMFP **Success rate** (%) on 2 3D domains.

| Domain | Task Descrip. | | Exec. Reason | | Exec. Result | | Goal Reach | | Average | | VLMFP | |
|---|---|---|---|---|---|---|---|---|---|---|---|---|
| | S | U | S | U | S | U | S | U | S | U | S | U |
| Assembly | 86.9 | 84.4 | 99.1 | 99.1 | 99.2 | 99.1 | 99.4 | 99.3 | 96.2 | 95.5 | 92.4 | 85.9 |
| MultiRob | 99.9 | 99.5 | 99.5 | 99.4 | 98.6 | 99.0 | 100.0 | 100.0 | 99.5 | 99.5 | 80.3 | 73.6 |

## 6 CONCLUSION

We introduce VLMFP, a Dual-VLM-guided framework that autonomously generates PDDL domain and problem files from visual observations for long-horizon visual planning. By combining a fine-tuned SimVLM for visual understanding and action simulation with a large GenVLM for symbolic reasoning and iterative refinement, VLMFP removes the need for pre-defined domain files, human supervision, or environment access. Experiments on six grid-world domains show that SimVLM reliably produces scenario descriptions and action simulations, while VLMFP consistently generates valid PDDL files enabling planners to solve diverse tasks. Results demonstrate strong generalization to unseen instances, appearances, and game rules, highlighting the value of integrating perception and symbolic reasoning for scalable formal planning. Beyond 2D benchmarks, we further show that VLMFP extends to complex long-horizon 3D planning tasks under partial observability and diverse visual variations, demonstrating the scalability of the perception-to-symbolic planning paradigm.

## 7 ETHICS STATEMENT

Our study uses only synthetic grid-world environments and does not involve human subjects, private information, or sensitive data. As such, there are no associated risks related to privacy, security, or legal compliance. While our framework advances automated planning by combining vision and symbolic reasoning, we acknowledge the potential misuse of planning systems in harmful applications. To mitigate such risks, our experiments are confined to safe benchmark domains, and we will release all code, models, and datasets to support transparency and reproducibility.

## 8 REPRODUCIBILITY STATEMENT

For reproducibility, we provide detailed domain descriptions in Appendix A.1, dataset statistics in Appendix B.5, fine-tuning parameters in Appendix B, and the details and prompts of VLMFP in Appendix C. The code is included in the Supplementary Material. Upon acceptance, we will release our code, model, and dataset publicly under an open-source license.

## 9 LARGE LANGUAGE MODEL USAGE FOR WRITING

In this paper, we employed large language models—specifically `ChatGPT`—solely as writing aids. Draft passages were submitted to the model to improve grammar and refine structure, and the results were carefully reviewed and edited by the authors. The use of the tool was strictly limited to text polishing; it was never applied to generate original content or references.

## 10 ACKNOWLEDGMENT

This work was supported by ONR under Award N00014-22-1-2478 and MIT-IBM Watson AI Lab. However, this article solely reflects the opinions and conclusions of its authors.

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

# SIMULATION TO RULES: A DUAL-VLM FRAMEWORK FOR FORMAL VISUAL PLANNING

# A  DOMAINS

## A.1  DOMAIN DESCRIPTIONS

We test on six grid world planning domains, **Frozenlake**, **Maze**, **Sokoban**, **Package**, **Printer**, and **Overcooked** (Towers et al., 2024; Silver & Chitnis, 2020; Jin et al., 2023; Wu et al., 2021):

- **Frozenlake** In the scenario, you have a girdworld, where each cell can be either normal ground or ice holes. The player starts at a cell, and there is a goal position in a cell. The goal is to move the player to the goal position. You can move up, down, left, right, but you cannot move into the border, and stepping into the ice hole will fail the game.

- **Maze** In the scenario, you have a maze gridworld with walls and goal positions. The player can move up, down, left, or right. If the player hits a wall or performs an invalid action, the action fails. The goal is to reach the goal cell.

- **Sokoban** In the scenario, you have a gridworld with walls, boxes, and goal positions. The player can move, push the box to goal, and push the box to other position. The player can only push the box forward, not toward other directions. If the player hits a wall or performs an invalid action, the action fails. The goal is to push all boxes to reach the goal cells.

- **Package** In the scenario, you have a gridworld with some closed packages in it. The player can turn-left, turn-right, move, pick-up, drop-down, open, or close the packages. If the player hits border or performs an invalid action, the action fails. The goal is to open all packages.

- **Printer** In the scenario, you have a gridworld with a desk region and a printer. The player can turn-left, turn-right, move, pick-up, drop-down, toggle-on, or toggle-off the printer. The player cannot move into the desk region. If the player hits border, desk, or performs an invalid action, the action fails. The goal is to pickup and drop the printer in desk region and toggle it on.

- **Overcooked** In the scenario, you have a gridworld cooking game with counters, ingredients, chopping boards, and a goal position for delivery. The players can move, chop, pick, drop, merge-ingredient, put-plate, deliver. If the player moves into the counter or performs an invalid action, the action fails. The goal is to cook the salad my merging chopped ingredients, put merged ingredients into plate, and put the plate to the delivery position.

We also test on 3D domains **Assembly** and **MultiRob** (Feng et al., 2025; Ji et al., 2025).

- **Assembly** In the scenario, there is a puzzle consisting of a board and several blocks with different colors on the table. The goal is to assemble the puzzle to reach the goal configuration with the robot arm. The robot arm can pick up, put down, insert, or reorient the block. The blocks have dependence on other blocks; that is, to achieve the goal, the blocks should be inserted after the blocks they depend on are inserted.

- **MultiRob** In the scenario, there is a a grid-like environment consisting of robot arms and objects with different colors on the table. The goal is to move all objects to their specified target locations. The robot arm can pickup and putdown the block. Each robot arm can only reach its surrounding four grids.

## A.2 SCENARIO APPEARANCE VISUALIZATIONS

We create six different scenario appearances for **Frozenlake**, **Maze**, **Sokoban**, **Package**, **Printer**, and five scenario appearances for **Overcooked**. We include the visualizations of different appearances in Fig. 5. For **Frozenlake**, **Maze**, **Sokoban**, we change the appearances for every object in the scenario, such as the frozen ice hole, the box, the wall, the agent, etc.. The change objects have various looks. For example, the agent can have the look of a person, a car, or a bike. And the goal can also have the look of a present, a home, a target, or a bag of money. For **Package** and **Printer**, we change the color of the background and the desk. For **Overcooked**, we not only change the color of the background, look of the agents and objects, but also change the ingredients from lettuce and tomato to tomato and onions. The differences in appearances not only bring challenges for the SimVLM to recognize unseen objects, but also to understand the game rule and reason about the objects in the scenario (lettuce or onion?).

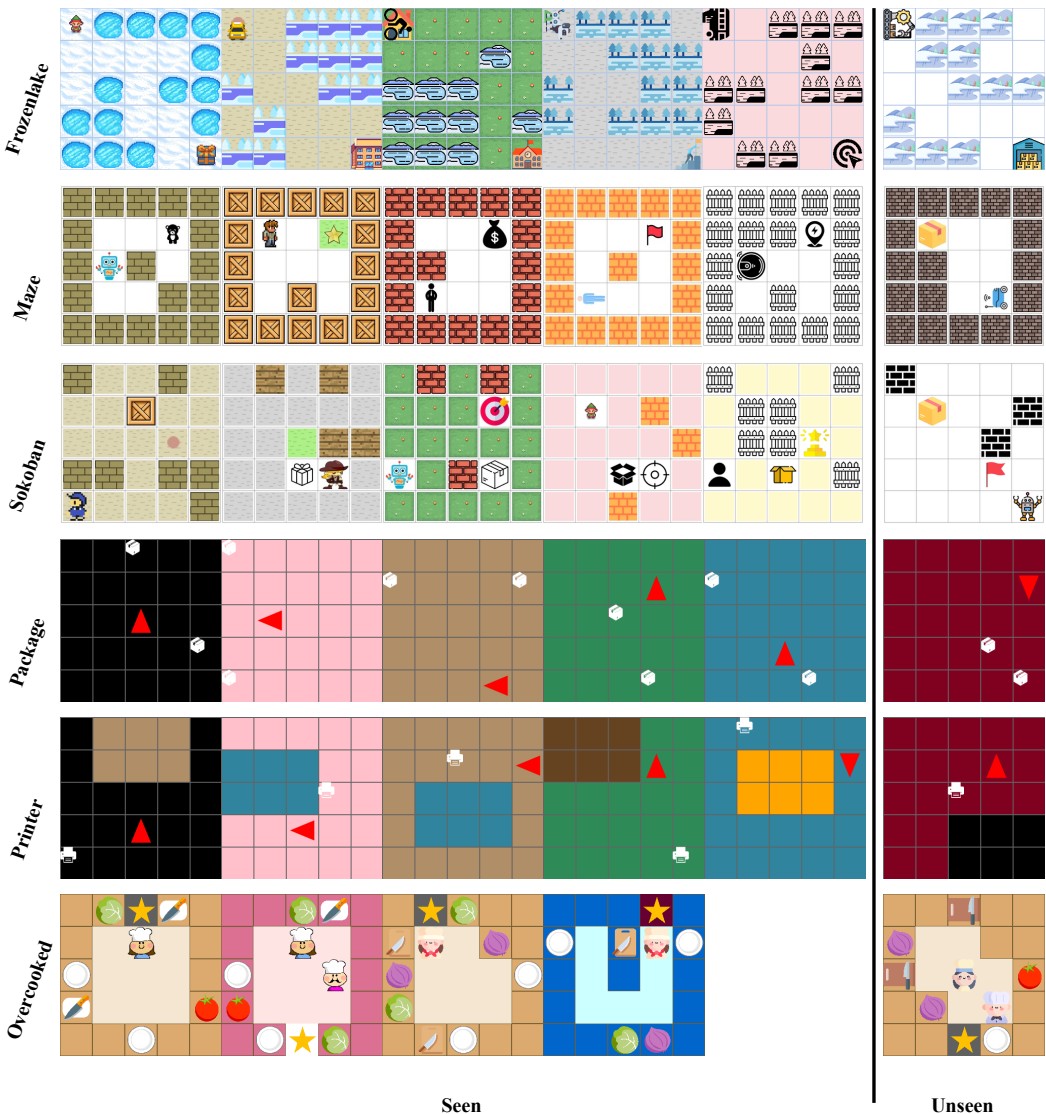

Figure 5: Full visualizations of various appearances for 6 grid world domains.

## A.3    DOMAIN COMPLEXITY COMPARISON

- **Frozenlake** 4 actions (move up, move down, move left, move right); Mechanism: no stepping into ice holes; Object types: grid positions
- **Maze** 4 actions (move up, move down, move left, move right); Mechanism: cannot move into walls; Object types: grid positions
- **Sokoban** 3 actions (move, push-to-goal, push-to-nongoal); Mechanism: movable objects; can only move/push to clear position; agent, box, and goal positions need to be three consecutive positions in one direction; Object types: grid positions, direction, box
- **Package** 5 actions (turn-left, turn-right, move, open, close); Mechanism: orientation; interactive objects; multiple objects to visit; Object types: grid positions, direction, package
- **Printer** 8 actions (turn-left, turn-right, move, pick, drop, drop_desk, toggle_on, toggle_off); Mechanism: orientation; interactive objects; Object types: grid positions, direction
- **Overcooked** 9 actions (move, pick-ingredient, drop-ingredient, pick-plate, drop-plate, chop, merge-ingredient, put-plate, deliver); Mechanism: multiple players; interactive objects; object different states; multiple objects to visit; Object types: grid positions, direction, player, ingredient, plate, chopping-board, delivery-point
- **Assembly** 6 actions (pickup, putdown, reorient_to_upright, reorient_to_horizontal, insert, unstack); Mechanism: 3D; object dependency; interactive objects; object different states; multiple objects to visit; no natural language goal description
- **MultiRob** 2 actions (pickup, putdown); Mechanism: 3D; multiple robots; occlusion; reachability; multiple perspectives; interactive objects; multiple objects to visit; Object types: grid positions, robot, object

When the domains become complex, without feedback and updating, it is easy for LLMs to ignore some preconditions/effects of actions or fail to assign every object a type, making it hard to directly generate correct domain and problem PDDL files. This significant performance difference showcases the importance of the feedback and updating modules when handling complex domains.

# B    SimVLM Details

## B.1    SimVLM baselines

We compare SimVLM with two baselines: 1) GPT4o-Direct: using GPT4o to direct accomplish the task, given the expected format; and 2) GPT4o-InContext: using GPT4o to accomplish the task, given the expected format and an in-context example. Note that we use the same four metrics (Task Description, Execution Reason, Execution Result, Goal Reach) for comparison. However, since for Execution Reason, we do not expect API-based VLMs to output exact formatted output as the ground truth output, we did not use exact string match. Instead, we use sentence-transformers library with all-MiniLM-L6-v2 model to check the Semantic Similarity for Execution Reason, which is a less strict evaluation than SimVLM.

The results are shown in Table 6. GPT4o-Direct and GPT4o-InContext can only accomplish the four tasks with an average of 2.5% and 3.3%(extremely low because this is not binary judgement), 51.6% and 76.5%(semantic similarity), 35.7% and 67.3%(binary judgement), 81.4% and 87.7%(binary judgement), comparing to 82.6%, 88.1%, 87.8%, 85.6% of SimVLM. We can observe that the API-based large VLM lacks precise spatial understanding and long-horizon reasoning capabilities to directly accomplish the task of SimVLM, thus fine-tuning a specialized model is necessary.

Table 6: **String matching rate** (%) comparison across four metrics on 6 grid world domains.

| Task Description | | | | | | | |
|---|---|---|---|---|---|---|---|
| Method | Frozenlake | Maze | Sokoban | Package | Printer | Overcooked | Average |
| Direct$_{GPT-4o}$ | 0.0 | 0.0 | 0.0 | 12.7 | 2.0 | 0.0 | 2.5 |
| InContext$_{GPT-4o}$ | 0.0 | 0.0 | 0.0 | 12.0 | 8.0 | 0.0 | 3.3 |
| **SimVLM** | 92.3 | 89.8 | 51.6 | 99.7 | 96.3 | 65.6 | 82.6 |
| **Execution Reason** | | | | | | | |
| Direct$_{GPT-4o}$ | 34.6 | 63.4 | 21.7 | 55.6 | 63.4 | 70.7 | 51.6 |
| InContext$_{GPT-4o}$ | 47.8 | 82.7 | 82.9 | 79.5 | 82.4 | 83.8 | 76.5 |
| **SimVLM** | 96.0 | 97.1 | 84.6 | 88.4 | 86.8 | 75.9 | 88.1 |
| **Execution Result** | | | | | | | |
| Direct$_{GPT-4o}$ | 29.1 | 42.9 | 16.9 | 23.9 | 56.6 | 44.8 | 35.7 |
| InContext$_{GPT-4o}$ | 32.1 | 63.0 | 73.4 | 82.8 | 83.6 | 68.9 | 67.3 |
| **SimVLM** | 95.9 | 96.8 | 83.4 | 88.4 | 86.8 | 75.7 | 87.8 |
| **Goal Reach** | | | | | | | |
| Direct$_{GPT-4o}$ | 72.0 | 68.8 | 80.8 | 79.2 | 91.4 | 96.0 | 81.4 |
| InContext$_{GPT-4o}$ | 83.8 | 74.4 | 96.8 | 83.8 | 89.0 | 99.2 | 87.7 |
| **SimVLM** | 93.7 | 94.7 | 83.2 | 86.0 | 84.6 | 71.2 | 85.6 |

## B.2 SimVLM base model ablation

We selected Qwen2-VL-7B (Wang et al., 2024) as our base model because it is one of the small VLMs that achieves state-of-the-art performance across a wide spectrum of multimodal benchmarks. We add an ablation study that evaluates SimVLM performance comparing Qwen2-VL-7B with LLaVA-NeXT-7B (Liu et al., 2024) and Google PaliGemma2-10B (Steiner et al., 2024). The results are shown in Table 7 and Table 8. Both models performed reasonably well, but they showed slightly lower accuracy, higher variance, and are less generalizable to unseen appearances than Qwen2-VL-7B. Importantly, although the choice of Qwen2-VL-7B reflects empirical stability, our framework does not depend on Qwen2-VL-7B specifically. VLMFP remains model-agnostic, and substituting the simulator backbone is straightforward.

Table 7: **String matching rate** (%) for 4 SimVLM output types on 6 grid world domains, with LLaVA-NeXT-7B as the base model.

| Output | Frozenlake | | Maze | | Sokoban | | Package | | Printer | | Overcooked | | Average | |
|---|---|---|---|---|---|---|---|---|---|---|---|---|---|---|
| | S | U | S | U | S | U | S | U | S | U | S | U | S | U |
| **Task Descrip.** | 99.3 | 87.2 | 99.5 | 30.7 | 85.9 | 61.2 | 99.0 | 98.6 | 99.7 | 80.7 | 97.0 | 43.0 | 96.7 | 66.9 |
| **Exec. Reason** | 89.2 | 86.2 | 74.5 | 91.2 | 64.2 | 81.2 | 79.5 | 79.2 | 75.3 | 75.7 | 68.5 | 68.5 | 75.2 | 80.3 |
| **Exec. Result** | 89.3 | 86.4 | 74.2 | 91.3 | 64.9 | 82.3 | 79.6 | 79.2 | 75.4 | 75.8 | 68.8 | 68.8 | 75.4 | 80.6 |
| **Goal Reach** | 89.6 | 87.6 | 73.9 | 91.2 | 65.9 | 83.5 | 80.1 | 79.7 | 75.9 | 76.3 | 69.2 | 69.2 | 75.8 | 81.2 |
| Average | 91.8 | 86.8 | 80.5 | 76.1 | 70.2 | 77.0 | 84.5 | 84.2 | 81.6 | 77.1 | 75.9 | 62.4 | 80.8 | 77.3 |

Table 8: **String matching rate** (%) for 4 SimVLM output types on 6 grid world domains, with PaliGemma2-10B as the base model.

| Output | Frozenlake | | Maze | | Sokoban | | Package | | Printer | | Overcooked | | Average | |
|---|---|---|---|---|---|---|---|---|---|---|---|---|---|---|
| | S | U | S | U | S | U | S | U | S | U | S | U | S | U |
| **Task Descrip.** | 99.8 | 88.5 | 99.6 | 13.2 | 76.7 | 50.6 | 95.2 | 94.4 | 88.1 | 85.8 | 99.7 | 42.5 | 93.2 | 62.5 |
| **Exec. Reason** | 90.9 | 87.2 | 77.5 | 94.0 | 68.5 | 84.7 | 83.0 | 82.8 | 78.9 | 78.8 | 72.5 | 72.8 | 78.6 | 83.4 |
| **Exec. Result** | 90.9 | 87.2 | 77.6 | 94.1 | 67.3 | 86.4 | 83.2 | 83.0 | 79.0 | 78.9 | 73.3 | 73.5 | 78.6 | 83.9 |
| **Goal Reach** | 90.9 | 89.7 | 77.2 | 94.0 | 71.0 | 87.2 | 83.1 | 82.8 | 80.0 | 79.7 | 74.1 | 74.1 | 79.4 | 84.6 |
| Average | 93.2 | 88.1 | 83.0 | 73.9 | 70.9 | 77.2 | 86.1 | 85.8 | 81.5 | 80.8 | 79.9 | 65.7 | 82.4 | 78.6 |

## B.3    SimVLM performance across different seeds

Without an external oracle, absolute correctness of SimVLM cannot be guaranteed. However, we conduct experiments to evaluate SimVLM across 3 random seeds to measure its stability. We show the mean and standard deviation of the string matching rates for 4 metrics across 3 seeds in Table 9 and Table 10. Across the 6 domains, the standard deviation for all four metrics is very small, indicating that SimVLM's perception, step-wise reasoning, and action-result predictions are highly stable and SimVLM does not hallucinate unpredictably.

Additionally, we emphasize that the SimVLM metrics are computed over 1000 evaluation datapoints, and VLMFP results are averaged over 1500 planning trials for each domain. These large-sample evaluations already demonstrate strong empirical stability despite the possibility of hallucination, and the resulting performance substantially exceeds all baseline models.

Table 9: **The mean of string matching rate** (%) for 4 SimVLM output types on 6 grid world domains across 3 seeds.

| Output | Frozenlake | | Maze | | Sokoban | | Package | | Printer | | Overcooked | | Average | |
|---|---|---|---|---|---|---|---|---|---|---|---|---|---|---|
| | S | U | S | U | S | U | S | U | S | U | S | U | S | U |
| **Task Descrip.** | 99.9 | 91.9 | 99.2 | 89.9 | 74.0 | 51.4 | 100.0 | 99.9 | 99.8 | 96.6 | 99.2 | 64.9 | 95.3 | 82.4 |
| **Exec. Reason** | 94.9 | 94.8 | 83.6 | 96.9 | 72.4 | 83.4 | 88.5 | 89.0 | 85.7 | 85.4 | 76.9 | 75.8 | 83.7 | 87.5 |
| **Exec. Result** | 94.9 | 94.7 | 83.6 | 97.0 | 72.1 | 83.0 | 88.5 | 89.0 | 85.7 | 85.4 | 77.0 | 75.9 | 83.7 | 87.5 |
| **Goal Reach** | 93.5 | 93.5 | 80.3 | 94.7 | 72.7 | 83.1 | 86.8 | 86.6 | 83.6 | 83.7 | 73.9 | 73.2 | 81.8 | 85.8 |
| Average | 95.8 | 93.7 | 86.7 | 94.6 | 72.8 | 75.2 | 91.0 | 91.1 | 88.7 | 87.8 | 81.8 | 72.4 | 86.1 | 85.8 |

Table 10: **The standard deviation of string matching rate** (%) for 4 SimVLM output types on 6 grid world domains across 3 seeds.

| Output | Frozenlake | | Maze | | Sokoban | | Package | | Printer | | Overcooked | | Average | |
|---|---|---|---|---|---|---|---|---|---|---|---|---|---|---|
| | S | U | S | U | S | U | S | U | S | U | S | U | S | U |
| **Task Descrip.** | 0.1 | 0.5 | 0.1 | 0.1 | 2.1 | 0.3 | 0.0 | 0.2 | 0.1 | 0.4 | 0.2 | 1.1 | 0.3 | 0.2 |
| **Exec. Reason** | 1.5 | 1.3 | 1.5 | 0.2 | 3.8 | 2.0 | 2.3 | 1.0 | 2.2 | 2.0 | 1.8 | 3.7 | 2.0 | 1.3 |
| **Exec. Result** | 1.4 | 1.3 | 1.5 | 0.2 | 3.6 | 0.6 | 2.3 | 1.0 | 2.2 | 2.0 | 1.6 | 3.7 | 2.0 | 1.0 |
| **Goal Reach** | 0.5 | 0.2 | 1.7 | 0.0 | 1.4 | 0.1 | 1.1 | 0.7 | 0.5 | 0.8 | 1.9 | 2.4 | 0.7 | 0.2 |
| Average | 0.8 | 0.7 | 0.7 | 0.0 | 2.2 | 0.6 | 1.4 | 0.7 | 1.2 | 1.1 | 0.9 | 2.5 | 1.1 | 0.6 |

### B.4 FAILURE CASE FOR GPT-5 TO ACCOMPLISH THE SAME TASK

For the following image observation as input, we show the GPT-5 generated task description and action execution results for the sequence [move left, move down, move down, move up, move up, move right]. From the output, we can observe that GPT-5 fails to recognize the locations of ice

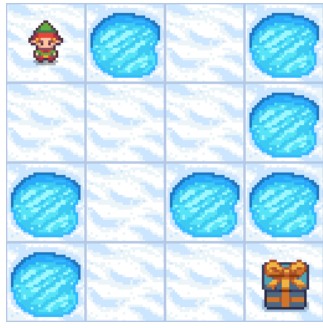

Figure 6: The input image to GPT-5 as the image observation of problem instance.

holes. In addition, since it mistakenly thinks pos-2-1 is an ice hole, the action execution also fails. The agent should successfully move to pos-2-1 on Step 2, while GPT-5 marks it as moving into the ice hole, which results in game failure. We include the correct output for this instance from SimVLM in Appendix B.7.

```
From the image we can observe the: Initial State Description:
Initially:
- The size of the gridworld is 4x4.
- The agent is at (pos-1-1).
- The goal is at (pos-4-4).
- The ice holes are at (pos-1-2), (pos-1-3), (pos-2-1), (pos-2-2), (pos
    -3-2), (pos-3-3).

Your goal is to achieve the following configuration:
- The agent is at (pos-4-4)

Step 1 - Action move left:
Execution Reasoning: The agent tries to move left from pos-1-1. This
    action is not valid because it would move into the border, so the
    agent stays at pos-1-1.
Execution result: Unsuccessful.
Step 2 - Action move down:
Execution Reasoning: The agent tries to move down from pos-1-1 to pos
    -2-1. Cell pos-2-1 is an ice hole, so the agent falls in and the game
     ends.
Execution result: Unsuccessful.
Step 3 - Action move down:
Execution Reasoning: The agent has already failed the game, so this
    execution is invalid.
Execution result: Invalid.
Step 4 - Action move up:
Execution Reasoning: The agent has already failed the game, so this
    execution is invalid.
Execution result: Invalid.
Step 5 - Action move up:
Execution Reasoning: The agent has already failed the game, so this
    execution is invalid.
Execution result: Invalid.
Step 6 - Action move right:
Execution Reasoning: The agent has already failed the game, so this
    execution is invalid.
Execution result: Invalid.
Goal reaching: Unsuccessful.
```

### B.5 DATASET STATISTICS

Table 11: Dataset and Task statistics for six grid world domains.

| Task | FrozenLake | Maze | Sokoban | Package | Printer | Overcooked |
|------|-----------|------|---------|---------|---------|------------|
| Grid Sizes | 3–8 | 5–8 | 5–8 | 4–8 | 4–8 | 5–8 |
| #Appearances (S/U) | 5/1 | 5/1 | 5/1 | 5/1 | 5/1 | 4/1 |
| Action Types | 4 | 4 | 3 | 7 | 7 | 7 |
| # Data per Appearances | 20000 | 20000 | 20000 | 5000 | 5000 | 19739 |

### B.6 FINE-TUNING IMPLEMENTATIONS

We utilize four H100 80GB GPUs for full-parameter fine-tuning of the Qwen2-VL-7B model. The model is trained for 3 epochs in the SFT stage. The learning rate is set to $1 \times 10^{-5}$. We use a batch size of 2 for training. We use 10% of the training data for validation, and we select the model with the lowest validation loss. We perform the inference of SimVLM using a single H100 80GB GPU.

### B.7 INPUT AND OUTPUT EXAMPLES

**Example Input** of the **Frozenlake** domain:

```
<image>Given a sequence of actions and an image observation of the
    initial setup of a scenario, your goal is to decide the execution
    result after executing given actions. To achieve this task, you will
    first describe the setup you observe from the image, give the
    reasoning that explains how the player moves and the success or
    failure reason for each single action, then rate whether the overall
    execution is successful, and eventually describe whether the actions
    achieves the task's goal.

Task Description: In the scenario, you have a girdworld, where each cell
    can be either normal ground or ice holes. The left upper corner is (
    pos-1-1) for (row, column) representation. The player starts at a
    cell, and there is a goal position in a cell. The goal is to move the
     player to the goal position. You can move up, down, left, right, but
     you cannot move into the border, and stepping into the ice hole will
     fail the game.
Action Sequence:
1: move left
2: move down
3: move down
4: move up
5: move up
6: move right
```

**Example Output** of the **Frozenlake** domain:

```
From the image we can observe the: Initial State Description:
Initially:
- The size of the gridworld is 4x4.
- The agent is at (pos-1-1).
- The goal is at (pos-4-4).
- The ice holes are at (pos-1-2), (pos-1-4), (pos-2-4), (pos-3-1), (pos
    -3-3), (pos-3-4), (pos-4-1).

Your goal is to achieve the following configuration:
- The agent is at (pos-4-4)

Step 1 - Action move left:
```

```
Execution Reasoning: The agent tries to move left from pos-1-1. This
    action is not valid as it is moving into the border. So the execution
     is unsuccessful and the agent stays at the original location.
Execution result: Unsuccessful.
----------------------------------------------
Step 2 - Action move down:
Execution Reasoning: The agent tries to move down from pos-1-1, so it
    will move to pos-2-1. There is no ice hole in cell pos-2-1 and the
    agent is not moving to the border, so no invalid action is executed
    and the agent successfully moves to pos-2-1.
Execution result: Successful.
----------------------------------------------
Step 3 - Action move down:
Execution Reasoning: The agent tries to move down from pos-2-1, so it
    will move to pos-3-1. Since there is an ice hole in cell pos-3-1, the
     agent will fall into the ice hole, so the game ended and the
    execution is unsuccessful.
Execution result: Unsuccessful.
----------------------------------------------
Step 4 - Action move up:
Execution Reasoning: The agent has already failed the game, so this
    execution is invalid. Execution result: Invalid.
----------------------------------------------
Step 5 - Action move up:
Execution Reasoning: The agent has already failed the game, so this
    execution is invalid. Execution result: Invalid.
----------------------------------------------
Step 6 - Action move right:
Execution Reasoning: The agent has already failed the game, so this
    execution is invalid. Execution result: Invalid.
----------------------------------------------
Goal reaching: Unsuccessful
```

**Example Input** of the **Assembly** domain:

```
<image><image>Given a sequence of actions, an image observation of the
    initial setup of a scenario, and an image observation of the goal
    configuration, your goal is to decide the execution result after
    executing given actions. To achieve this task, you will first
    describe the setup and goal you observe from the image, give the
    reasoning that explains how the agent moves and the success or
    failure reason for each single action, then rate whether the overall
    execution is successful, and eventually describe whether the actions
    achieves the task's goal.

Task Description: In the scenario, There is a puzzle consisting of a
    board and several blocks with different colors on the table. The goal
     is to assemble the puzzle with the robot arm. The robot arm can
    pickup, putdown, insert, or reorient the block. If the robot arm
    performs an invalid action, the action fails. The goal is to assembly
     the blocks to reach the goal configuration. In the goal
    configuration, some blocks have dependence to other blocks, that is,
    to achieve the goal, the blocks should be inserted after the blocks
    they depend on are inserted.
Action Sequence:
1: reorient red nail
2: insert blue block
3: pick up red nail
4: insert yellow block
```

**Example Output** of the **Assembly** domain:

```
From the images we can observe that:
Initially:
```

```
- There are 7 blocks.
- The blue block is upright; it is not inserted.
- The brown block is upright; it is not inserted.
- The orange block is upright; it is inserted.
- The purple block is upright; it is not inserted.
- The yellow block is not upright; it is not inserted.
- The pink nail is upright; it is inserted.
- The red nail is upright; it is not inserted.

- The blue block depends on orange block and purple block.
- The brown block depends on blue block and yellow block.
- The yellow block depends on orange block and purple block.
- The pink nail depends on brown block.
- The red nail depends on orange block.

Your goal is to achieve the following configuration:
- All blocks are inserted.

Step 1 - Action reorient red nail:
Execution Reasoning: The robot arm tries to reorient red nail, but it
    cannot reorient object red nail since it is not in hand.
Execution result: Unsuccessful.
----------------------------------------------
Step 2 - Action insert blue block:
Execution Reasoning: The robot arm tries to insert blue block, but it
    cannot insert object blue block since it is not in hand.
Execution result: Unsuccessful.
----------------------------------------------
Step 3 - Action pick up red nail:
Execution Reasoning: The robot arm tries to pick up red nail. Since no
    invalid action is executed, the agent successfully pick up red nail.
Execution result: Successful.
----------------------------------------------
Step 4 - Action insert yellow block:
Execution Reasoning: The robot arm tries to insert yellow block, but it
    cannot insert object yellow block since it is not in hand.
Execution result: Unsuccessful.
----------------------------------------------
Goal reaching: Unsuccessful
```

## C VLMFP DETAILS

### C.1 DIRECT AND COT MAJOR FAILURE CASES

For all domains, one major failure case for Direct and CoT is because the failure to correctly recognize and locate the objects in the scenario due to insufficient image recognition and spatial reasoning capability. In addition to this, some other failures result from the misunderstanding of actions. For example, in the Package, Printer, and Overcooked domain, when the agent stands in a cell, it can perform actions, such as open and close, to objects in adjacent cells. But the VLMs often think the agent should stand in the same cell as the object to operate it. Including in-context examples is beneficial, but could not stop all failures like this.

### C.2 VLMFP MAJOR FAILURE MODES AND CASES

SimVLM perception errors and GenVLM generation errors are the major failure modes. We looked into the failure reasons of the 15 input instances for both seen and unseen appearances for all domains and summarized them in Table 13.

Table 12: Error type distribution of VLMFP for 15 input instances on 6 domains. Values show errors/total_failures for each category.

| Error Type | Frozenlake | | Maze | | Sokoban | | Package | | Printer | | Overcooked | |
|---|---|---|---|---|---|---|---|---|---|---|---|---|
| | S | U | S | U | S | U | S | U | S | U | S | U |
| Perception | 0/0 | 0/1 | 0/1 | 1/3 | 2/5 | 5/8 | 0/4 | 0/7 | 0/5 | 0/5 | 0/8 | 5/10 |
| Generation | 0/0 | 1/1 | 1/1 | 2/3 | 3/5 | 3/8 | 4/4 | 7/7 | 5/5 | 5/5 | 8/8 | 5/10 |

From the table, we can observe that for visually busier scenes(sokoban) and scenarios with unseen objects (onions in Overcooked U), the perception error becomes more common. While in other cases where the domain mechanisms are complex with various constraints, failing to generate and update correct action specifications, necessary predicates, or initializing objects within limited rounds is the major failure case.

More specifically, for failure in generating correct PDDL problem files, one major failure case for all domains is that it fails to generate all states that describe directional relationships between cells. For example, 'move-dir pos-1-1 pos-1-2 right' defines 'pos-1-2 is at right to pos-1-1'. Although the problem description is correct, the GenVLM sometimes fails to include all available directional relationships, so that some actions cannot be performed. In addition to this type of failure, for complex domains, there are often different types of objects. Sometimes, GenVLM fails to assign types for all objects.

For failure in generating correct Domain problem files, the major failure case is failure to describe correct action preconditions and effects. For example, in the Package domain, the agent can only open the package it faces. The definition for the open action is:

```
(:action open
  :parameters (?pkg - package ?pos - position ?pkgpos - position ?dir -
      direction)
  :precondition (and
    (at ?pos)
    (package-at ?pkg ?pkgpos)
    (package-closed ?pkg)
    (facing ?dir)
    (move-dir ?pos ?pkgpos ?dir)
  )
  :effect (and
    (not (package-closed ?pkg))
    (package-open ?pkg)
  )
)
```

This 'facing ?dir' precondition and the '(move-dir ?pos ?pkgpos ?dir)' are easy to neglect.

For CodePDDL, it has similar failure cases as VLMFP. Without the updating process, the failures cannot be corrected.

We believe that more techniques could be implemented to mitigate both perception errors and generation errors. We observe during development of VLMFP that the generalization capability of SimVLM improved as it sees more diverse domains, map sizes, appearances, and rules. Thus, there is great potential to develop a more generalized simulation 'world model' when it exposes stronger VLM base models to increasingly diverse environments. Additionally, the errors of GenVLM could decrease with more PDDL-targeted prompting and checkers, more updating rounds, and stronger VLM reasoning capabilities.

There remain many promising directions, both engineering-oriented and research-oriented, that could further strengthen the framework. We leave these explorations to future work and emphasize that VLMFP is, to the best of our knowledge, the first framework to generate both PDDL domain and problem files directly from visual inputs, without human feedback or environment access. We believe that establishing a reliable and generalizable visual-to-symbolic planning pipeline is a substantial contribution to VLM-based long-horizon planning and opens numerous avenues for future research.

## C.3 VLMFP IMPLEMENTATION DETAILS AND PROMPTS

**Prescreening** During prescreening, GenVLM first generates the problem PDDL file based on the domain description, problem description, and image observation. The descriptions include the problem and domain PDDL template, which only includes the available object types, action names, and action variable names. In the prompt, a non-task-specific example is included. The example is a 3x3 grid world with no obstacles. The prompt is:

```
Given a natural language description of a planning problem and an image
    observation of the initial scenario, your task is to generate a
    complete PDDL problem instance that is equivalent to its natural
    language description and image observation, contain all necessary
    predicates as described in the description, and is thorough and
    complete. Consider all predicates that are relevant and neccesary.
Example:
Domain Description:
{domain_nl_wrapped}
Domain PDDL Template:
```pddl
(define (domain grid)
  (:requirements :strips)
  (:predicates)

  (:action move-up
    :parameters (?from ?to)
    :precondition ()
    :effect ()
  )

  (:action move-down
    :parameters (?from ?to)
    :precondition ()
    :effect ()
  )

  (:action move-left
    :parameters (?from ?to)
    :precondition ()
    :effect ()
  )

  (:action move-right
    :parameters (?from ?to)
```

```
      :precondition ()
      :effect ()
  )
)
```
Problem Description:
```markdown
You are tasked with manipulating an agent to reach the goal without
    colliding into the wall. The position representation is (row, column)
     representation. For example, (pos-4-3) represents the position in
    fourth row and third column. The left upper corner is (pos-1-1). You
    can perform four actions: move-up, move-down, move-left, and move-
    right.

Initially:
- The agent is at (pos-2-1).
- The goal is at (pos-3-3).

Your goal is to achieve the following configuration:
- The agent is at (pos-3-3)
```

Problem PDDL Template:
```pddl
(define (problem grid)
    (:domain grid)
    (:objects )
    (:init )
    (:goal (and ))
)
```
Problem PDDL:
```markdown

(define (problem grid) (:domain grid)
  (:objects
        pos-1-1 - position
        pos-1-2 - position
        pos-1-3 - position
        pos-2-1 - position
        pos-2-2 - position
        pos-2-3 - position
        pos-3-1 - position
        pos-3-2 - position
        pos-3-3 - position
  )
  (:init
    (at pos-2-1)
    (is-goal pos-3-3)
    (move-dir-left pos-1-2 pos-1-1)
    (move-dir-left pos-1-3 pos-1-2)
    (move-dir-left pos-2-2 pos-2-1)
    (move-dir-left pos-2-3 pos-2-2)
    (move-dir-left pos-3-2 pos-3-1)
    (move-dir-left pos-3-3 pos-3-2)
    (move-dir-right pos-1-1 pos-1-2)
    (move-dir-right pos-1-2 pos-1-3)
    (move-dir-right pos-2-1 pos-2-2)
    (move-dir-right pos-2-2 pos-2-3)
    (move-dir-right pos-3-1 pos-3-2)
    (move-dir-right pos-3-2 pos-3-3)
    (move-dir-up pos-2-1 pos-1-1)
    (move-dir-up pos-2-2 pos-1-2)
    (move-dir-up pos-2-3 pos-1-3)
    (move-dir-up pos-3-1 pos-2-1)
```

```
      (move-dir-up pos-3-2 pos-2-2)
      (move-dir-up pos-3-3 pos-2-3)
      (move-dir-down pos-1-1 pos-2-1)
      (move-dir-down pos-1-2 pos-2-2)
      (move-dir-down pos-1-3 pos-2-3)
      (move-dir-down pos-2-1 pos-3-1)
      (move-dir-down pos-2-2 pos-3-2)
      (move-dir-down pos-2-3 pos-3-3)
   )
   (:goal (at pos-3-2))
)
```
Domain Description:
{target_domain_nl}
Domain PDDL Template:
{target_domain_template_pddl}

Problem Description:
{target_problem_nl}

Problem PDDL Template:
{target_problem_template_pddl}
```

After generating the problem file, GenVLM continues to generate the domain file based on the example of the grid, domain, and problem descriptions, image observation, and the generated problem file. The prompt for this is:

```
You are given a natural language description of a planning problem in the
    domain {target_domain_name} along with one problem instance in PDDL
    format and an image observation of the initial setup. Your task is to
    generate a PDDL domain for the target domain {target_domain_name}
    that is equivalent to its natural language description and is
    compatible with the provided problem instance.

Starting from a PDDL domain template, you are allowed to modify the
    template using the following two python function interfaces:

```python
add_or_update_predicates(predicates: List[str])
modify_action(action_name: str, new_preconditions: List[str], new_effects
    : List[str])
```

An example of above functions applied to an example PDDL domain template
    is as follows:

Example Domain Description:
```markdown
The robot has four actions: move-up, move-down, move-left, and move-right
    . This domain models an agent navigating in a grid world. The goal is
     to reach a target location. The world consists of a set of discrete
    positions (e.g., grid cells). The position representation is (row,
    column) representation. For example, (pos-4-3) represents the
    position in fourth row and third column, and (pos-4-4) is at right of
     (pos-4-3). Between each two positions, the directional links between
     them design how they can move between each other. For example, the
    agent can move-left from pos-3-2 to pos-3-1 because pos-3-1 is at
    left to pos-3-2, but cannot move-left from pos-3-2 to pos-3-3.
```

Example Problem PDDL:
```pddl

(define (problem maze) (:domain maze)
```

```
  (:objects
        pos-1-1 - position
        pos-1-2 - position
        pos-1-3 - position
        pos-2-1 - position
        pos-2-2 - position
        pos-2-3 - position
        pos-3-1 - position
        pos-3-2 - position
        pos-3-3 - position
  )
  (:init
    (at pos-2-1)
    (is-goal pos-3-3)
    (move-dir-left pos-1-2 pos-1-1)
    (move-dir-left pos-1-3 pos-1-2)
    (move-dir-left pos-2-2 pos-2-1)
    (move-dir-left pos-2-3 pos-2-2)
    (move-dir-left pos-3-2 pos-3-1)
    (move-dir-left pos-3-3 pos-3-2)
    (move-dir-right pos-1-1 pos-1-2)
    (move-dir-right pos-1-2 pos-1-3)
    (move-dir-right pos-2-1 pos-2-2)
    (move-dir-right pos-2-2 pos-2-3)
    (move-dir-right pos-3-1 pos-3-2)
    (move-dir-right pos-3-2 pos-3-3)
    (move-dir-up pos-2-1 pos-1-1)
    (move-dir-up pos-2-2 pos-1-2)
    (move-dir-up pos-2-3 pos-1-3)
    (move-dir-up pos-3-1 pos-2-1)
    (move-dir-up pos-3-2 pos-2-2)
    (move-dir-up pos-3-3 pos-2-3)
    (move-dir-down pos-1-1 pos-2-1)
    (move-dir-down pos-1-2 pos-2-2)
    (move-dir-down pos-1-3 pos-2-3)
    (move-dir-down pos-2-1 pos-3-1)
    (move-dir-down pos-2-2 pos-3-2)
    (move-dir-down pos-2-3 pos-3-3)
  )
   (:goal (at pos-3-2))
)
```

Example PDDL Template:
```pddl
(define (problem maze)
    (:domain maze)
    (:objects )
    (:init )
    (:goal (and ))
)
```

Example Completion:
```python
add_or_update_predicates([
    '(move-dir-up ?x ?y)',
    '(move-dir-down ?x ?y)',
    '(move-dir-left ?x ?y)',
    '(move-dir-right ?x ?y)',
    '(at ?x)',
    '(is-goal ?x)'
])
modify_action('move-up', [
    "(at ?from)",
```

```
        "(move-dir-up ?from ?to)"
], [
    "(not (at ?from))",
    "(not (clear ?to))",
    "(at ?to)",
])

modify_action('move-down', [
    "(at ?from)",
    "(move-dir-down ?from ?to)"
], [
    "(not (at ?from))",
    "(at ?to)",
])

modify_action('move-left', [
    "(at ?from)",
    "(move-dir-left ?from ?to)"
], [
    "(not (at ?from))",
    "(at ?to)",
])

modify_action('move-right', [
    "(at ?from)",
    "(move-dir-right ?from ?to)"
], [
    "(not (at ?from))",
    "(at ?to)",
])
```

Target Domain Description:
{target_domain_nl}

Target Problem PDDL:
{target_problem_pddl}

Now, your task is to complete the following PDDL template by generating
    necessary predicates and action preconditions and effects:

Target PDDL Template:
{target_domain_template_pddl}

You must never modify action parameters, and you are only allowed to use
    the following two function interfaces to modify the template.
```

**PDDL files updating** During updating, GenVLM takes in the descriptions, image observation, previously generated problem and domain PDDL files, and execution inconsistency feedback. GenVLM is prompted to reason about which part of the files is problematic and then update the file(s). The prompt for this is:

```
Here are your generated PDDL problem and domain. Please reason about the
    issue with your generated problem file or generated code for domain
    generation.
Problem Description:
{target_problem_nl}

Domain Description:
{target_domain_nl}

Domain PDDL Template:
{target_domain_template}
```

```
Generated Problem PDDL:
{problem_pddl}

Generated Domain PDDL:
{domain_pddl}

Execution Feedback:
{execution_feedback}

In your response, please
1. reason about if you want to update your problem file and domain file
   to fix the issue by answering
Q1: List all predicates. Describe how they are related to objects and
    actions and how they should be updated.
Q2: List all actions in domain. Describe, explain, and compare action's
    definition in given natural language description with its inputs,
    preconditions, and effects in domain file one by one. No two actions
    should have same preconditions and effects. If not, please correct in
     domain file.
Q3: List all objects in problem.
Q4: Check the :objects section. If fixed 'typing' exists, list the type
    of each object and if it is EXPLICITLY written as format [object_name
     - type] in problem file. Explicit typing means the exact string - <
    type> must appear after each object name. Do not assume implicit
    typing. If any object is listed without a type, then the problem file
     is incorrect and must be modified.
Q5: List all init states in problem. Predicates used in domain should be
    initialized in problem file. And they should describe every aspect of
     the problem, and be enough to execute actions. When defining
    directional links of a grid, the init states should exhaustively
    enumerate all adjacent positions for every applicable cell in this
    grid, from first row and column to last row and column, no matter it
    is valid movement or contains obstacles or not. If not, please
    correct in problem file.
Q6: List all goals in problem.
Q7: Based on previous answers, summarize the errors one by one, and
    explain which file(s) should be updated and which parts of them
    should be updated in detail.

2. update the problem file to be correct and complete according to your
    analysis, if needed (Your modified PDDL will be directly used, so
    return only complete and valid PDDL: no comments, TODOs, placeholders
    , or omissions);

3. generate the new code for domain generation according to your analysis
    , if needed.

If you want to modify the new domain file, you are only allowed to use
    the following two function interfaces to modify the template.

```python
add_or_update_predicates(predicates: List[str])
modify_action(action_name: str, new_preconditions: List[str], new_effects
    : List[str])
```

An example of above functions is as follows:

```python
add_or_update_predicates(['(is-robot ?x)'])
modify_action('move', ['(is-robot ?x)'], ['(is-robot ?y)'])
```

Please give your answer strictly in the following format:
```

```
Problem file and Domain file update reasoning: []
New problem file: [fill in N/A if not needed]
New domain file: [fill in N/A if not needed]"
"""
```

The updating loop is skipped when the EW score reaches 1.0 ($abs(ew\_rating - 1.0) < 1e^{-6}$) and the plan generated using current PDDL files is valid. This means, the generated PDDL files are returned when they are verified by SimVLM to be able to solve the current instance and can achieve the same results as SimVLM for all explorations.

The iteration repeats for 4 times, and if EW score cannot achieve 1.0 or the generated plan is not valid, the generated PDDL files are returned. However, this does not necessarily imply that the generated files are entirely incorrect. For example, when the problem PDDL file is incorrect but the domain PDDL is correct, a well-specified domain PDDL generalizes across instances, enabling potential successful solutions for other instances.

We collected the convergence rates for all 6 domains, each is evaluated on 15 problem instances, with the maximal round of 4. On average, 77.1% and 64.8% problem instances are converged for seen and unseen appearances, respectively. These results show that the EW-guided refinement loop is effective across diverse domains and appearance variations. We find that convergence tends to be fastest in domains with simpler transition rules (e.g., FrozenLake, Maze) and more gradual in domains with richer multi-object interactions (e.g., Sokoban, Overcooked), which is consistent with our failure-mode analysis in Appendix C.2.

Table 13: Convergence Rate (%) of VLMFP for 15 input instances on 6 domains.

| Convergence | Frozenlake | | Maze | | Sokoban | | Package | | Printer | | Overcooked | | Average | |
|---|---|---|---|---|---|---|---|---|---|---|---|---|---|---|
| | S | U | S | U | S | U | S | U | S | U | S | U | S | U |
| VLMFP | 100.0 | 86.7 | 93.3 | 80.0 | 66.7 | 33.3 | 73.3 | 53.3 | 66.7 | 66.7 | 40.0 | 33.3 | 77.1 | 64.8 |

## D   GENELIZATION TO GAME RULES

### D.1   GAME RULE VARIATION DESCRIPTIONS

We designed 15 new rules and collected data for these rules under the default Frozenlake appearance, with size 3-8, and different ice hole probabilities. For each rule, we collect 20k data. Here's the detailed rule descriptions for the new rules.

- **R1:** Ice holes do not result in failure. Instead, the agent must step on one ice hole once as a precondition for game completion.
- **R2:** Variant of R1. The agent must step on two ice holes as a precondition for game completion.
- **R3:** Ice holes function as teleports. There are two ice holes in the scenario that allow teleportation between each other.
- **R4:** Ice holes function as teleports. Any ice hole allows the agent to teleport back to the origin position.
- **R5:** If the agent steps on an ice hole, it must execute the same action twice to actually execute it.
- **R6:** The agent has two lives. It does not fail when stepping on the first ice hole, but fails the game when stepping on the second one.
- **R7:** Stepping on an ice hole unlocks a lucky rocket, allowing the agent to step forward two steps in the same direction.
- **R8:** Ice is slippery. If the agent steps on ice and the next cell in the same direction is also ice, the agent slips to the second ice position, continuing to slip until reaching a non-ice position.
- **R9:** The agent can only step into an ice hole if entering from above. Stepping in from other directions is invalid.
- **R10:** After stepping on an ice hole, the agent must always execute actions twice to actually execute them.
- **R11:** Variant of R10. The agent must execute actions three times instead of twice.
- **R12:** If the agent steps onto ice, it slides in that direction until hitting a wall.
- **R13:** Variant of R12. The agent bounces back until reaching a wall instead of sliding forward.
- **R14:** Stepping on an ice hole swaps the goal and origin positions.
- **R15:** The agent can only move up or down when on ice holes.

We test on 5 unseen rules:

- **Rule1:** Since ice is wet, stepping on an ice hole causes the agent to step forward two steps in the same direction.
- **Rule2:** Ice holes are teleports to pos-2-2;
- **Rule3:** If you step on an ice hole, you have to execute the same actions three times to actually execute it;
- **Rule4:** Stepping on an ice hole unlocks a lucky rocket, where you can step forward three steps in the same direction;
- **Rule5:** If you step on an ice hole, you freeze and your next action is skipped.

# E  GROUND TRUTH PDDL FILES EXAMPLES

Here we show ground truth PDDL Problem and Domain file examples for the Frozenlake and Package domain. They represent very different sets of actions in grid world domains. In Frozenlake, the agent does not have orientations; that is, it can freely move to any adjacent cell. However, in the Package domain, the agent has orientation and can only move in the direction it faces. Thus, the move action definition is different, and the turn action is also required in the Package domain. In addition, the Package domain also shows some manipulation actions, which are common in complex grid world domains.

**Frozenlake Sample PDDL Problem File**:

```
(define (problem FL-rand)
    (:domain frozenlake)
    (:objects pos-1-1 pos-1-2 pos-1-3 pos-1-4 pos-2-1 pos-2-2 pos-2-3 pos
        -2-4 pos-3-1 pos-3-2 pos-3-3 pos-3-4 pos-4-1 pos-4-2 pos-4-3 pos
        -4-4)
    (:init
    (at pos-1-1)
    (ice-hole pos-1-3)
    (ice-hole pos-1-4)
    (ice-hole pos-2-2)
    (ice-hole pos-3-3)
    (ice-hole pos-3-4)
    (ice-hole pos-4-1)
    (up_direction pos-2-1 pos-1-1)
    (up_direction pos-2-2 pos-1-2)
    (up_direction pos-2-3 pos-1-3)
    (up_direction pos-2-4 pos-1-4)
    (up_direction pos-3-1 pos-2-1)
    (up_direction pos-3-2 pos-2-2)
    (up_direction pos-3-3 pos-2-3)
    (up_direction pos-3-4 pos-2-4)
    (up_direction pos-4-1 pos-3-1)
    (up_direction pos-4-2 pos-3-2)
    (up_direction pos-4-3 pos-3-3)
    (up_direction pos-4-4 pos-3-4)
    (down_direction pos-1-1 pos-2-1)
    (down_direction pos-1-2 pos-2-2)
    (down_direction pos-1-3 pos-2-3)
    (down_direction pos-1-4 pos-2-4)
    (down_direction pos-2-1 pos-3-1)
    (down_direction pos-2-2 pos-3-2)
    (down_direction pos-2-3 pos-3-3)
    (down_direction pos-2-4 pos-3-4)
    (down_direction pos-3-1 pos-4-1)
    (down_direction pos-3-2 pos-4-2)
    (down_direction pos-3-3 pos-4-3)
    (down_direction pos-3-4 pos-4-4)
    (left_direction pos-1-2 pos-1-1)
    (left_direction pos-1-3 pos-1-2)
    (left_direction pos-1-4 pos-1-3)
    (left_direction pos-2-2 pos-2-1)
    (left_direction pos-2-3 pos-2-2)
    (left_direction pos-2-4 pos-2-3)
    (left_direction pos-3-2 pos-3-1)
    (left_direction pos-3-3 pos-3-2)
    (left_direction pos-3-4 pos-3-3)
    (left_direction pos-4-2 pos-4-1)
    (left_direction pos-4-3 pos-4-2)
    (left_direction pos-4-4 pos-4-3)
    (right_direction pos-1-1 pos-1-2)
    (right_direction pos-1-2 pos-1-3)
    (right_direction pos-1-3 pos-1-4)
```

```
    (right_direction pos-2-1 pos-2-2)
    (right_direction pos-2-2 pos-2-3)
    (right_direction pos-2-3 pos-2-4)
    (right_direction pos-3-1 pos-3-2)
    (right_direction pos-3-2 pos-3-3)
    (right_direction pos-3-3 pos-3-4)
    (right_direction pos-4-1 pos-4-2)
    (right_direction pos-4-2 pos-4-3)
    (right_direction pos-4-3 pos-4-4)
    )
    (:goal
    (and
        (at pos-4-4)
    )
)
)
```

**Frozenlake PDDL Domain File**:

```
(define (domain frozenlake)
  (:requirements :strips)
  (:predicates (at ?x)  (down_direction ?from ?to)  (ice-hole ?x)  (
      left_direction ?from ?to)  (right_direction ?from ?to)  (
      up_direction ?from ?to))
    (:action move-down
        :parameters (?from ?to)
        :precondition (and (at ?from) (down_direction ?from ?to) (not (
            ice-hole ?from)))
        :effect (and (at ?to) (not (at ?from)))
    )
     (:action move-left
        :parameters (?from ?to)
        :precondition (and (at ?from) (left_direction ?from ?to) (not (
            ice-hole ?from)))
        :effect (and (at ?to) (not (at ?from)))
    )
     (:action move-right
        :parameters (?from ?to)
        :precondition (and (at ?from) (right_direction ?from ?to) (not (
            ice-hole ?from)))
        :effect (and (at ?to) (not (at ?from)))
    )
     (:action move-up
        :parameters (?from ?to)
        :precondition (and (at ?from) (up_direction ?from ?to) (not (ice-
            hole ?from)))
        :effect (and (at ?to) (not (at ?from)))
    )

)
```

**Package Sample PDDL Problem File**:

```
(define (problem package)
  (:domain package)

  (:objects
    ; Positions in 4x4 grid
    pos-1-1 pos-1-2 pos-1-3 pos-1-4 pos-2-1 pos-2-2 pos-2-3 pos-2-4 pos
        -3-1 pos-3-2 pos-3-3 pos-3-4 pos-4-1 pos-4-2 pos-4-3 pos-4-4 -
        position

    ; Packages
    pkg-1 pkg-2 - package
```

```
    ; Directions
    up down left right - direction
  )

  (:init
    ; Agent initial position and orientation
    (at pos-3-3)
    (facing up)

    ; Package locations and states
    (package-at pkg-1 pos-1-3)
    (package-closed pkg-1)
    (package-at pkg-2 pos-4-1)
    (package-closed pkg-2)

    ; Turn relations
    (left-turn up left)
    (left-turn left down)
    (left-turn down right)
    (left-turn right up)

    (right-turn up right)
    (right-turn right down)
    (right-turn down left)
    (right-turn left up)

    ; Grid adjacency relationships
    (move-dir pos-1-1 pos-1-2 right) (move-dir pos-1-2 pos-1-1 left)
    (move-dir pos-1-2 pos-1-3 right) (move-dir pos-1-3 pos-1-2 left)
    (move-dir pos-1-3 pos-1-4 right) (move-dir pos-1-4 pos-1-3 left)
    (move-dir pos-2-1 pos-2-2 right) (move-dir pos-2-2 pos-2-1 left)
    (move-dir pos-2-2 pos-2-3 right) (move-dir pos-2-3 pos-2-2 left)
    (move-dir pos-2-3 pos-2-4 right) (move-dir pos-2-4 pos-2-3 left)
    (move-dir pos-3-1 pos-3-2 right) (move-dir pos-3-2 pos-3-1 left)
    (move-dir pos-3-2 pos-3-3 right) (move-dir pos-3-3 pos-3-2 left)
    (move-dir pos-3-3 pos-3-4 right) (move-dir pos-3-4 pos-3-3 left)
    (move-dir pos-4-1 pos-4-2 right) (move-dir pos-4-2 pos-4-1 left)
    (move-dir pos-4-2 pos-4-3 right) (move-dir pos-4-3 pos-4-2 left)
    (move-dir pos-4-3 pos-4-4 right) (move-dir pos-4-4 pos-4-3 left)
    (move-dir pos-1-1 pos-2-1 down) (move-dir pos-2-1 pos-1-1 up)
    (move-dir pos-1-2 pos-2-2 down) (move-dir pos-2-2 pos-1-2 up)
    (move-dir pos-1-3 pos-2-3 down) (move-dir pos-2-3 pos-1-3 up)
    (move-dir pos-1-4 pos-2-4 down) (move-dir pos-2-4 pos-1-4 up)
    (move-dir pos-2-1 pos-3-1 down) (move-dir pos-3-1 pos-2-1 up)
    (move-dir pos-2-2 pos-3-2 down) (move-dir pos-3-2 pos-2-2 up)
    (move-dir pos-2-3 pos-3-3 down) (move-dir pos-3-3 pos-2-3 up)
    (move-dir pos-2-4 pos-3-4 down) (move-dir pos-3-4 pos-2-4 up)
    (move-dir pos-3-1 pos-4-1 down) (move-dir pos-4-1 pos-3-1 up)
    (move-dir pos-3-2 pos-4-2 down) (move-dir pos-4-2 pos-3-2 up)
    (move-dir pos-3-3 pos-4-3 down) (move-dir pos-4-3 pos-3-3 up)
    (move-dir pos-3-4 pos-4-4 down) (move-dir pos-4-4 pos-3-4 up)
  )

  (:goal
    (and
      (package-open pkg-1)
      (package-open pkg-2)
    )
  )
)
```

**Package PDDL Domain File**:

```
(define (domain package)
  (:requirements :strips :typing)

  (:types
    position  package  direction
  )

  (:predicates
    (at ?pos - position)                ; agent is at position
    (package-at ?pkg - package ?pos - position) ; package is at position
    (package-open ?pkg - package)     ; package is open
    (package-closed ?pkg - package)   ; package is closed
    (facing ?dir - direction)           ; agent is facing direction
    (left-turn ?from - direction ?to - direction)
    (right-turn ?from - direction ?to - direction)
    (move-dir ?pos1 - position ?pos2 - position ?dir - direction) ;
        positions are move-dir in direction
  )

  (:action turn-left
    :parameters (?current-dir - direction ?new-dir - direction)
    :precondition (and
      (facing ?current-dir)
      (left-turn ?current-dir ?new-dir)
    )
    :effect (and
      (not (facing ?current-dir))
      (facing ?new-dir)
    )
  )

  (:action turn-right
    :parameters (?current-dir - direction ?new-dir - direction)
    :precondition (and
      (facing ?current-dir)
      (right-turn ?current-dir ?new-dir)
    )
    :effect (and
      (not (facing ?current-dir))
      (facing ?new-dir)
    )
  )

  (:action move
    :parameters (?from - position ?to - position ?dir - direction)
    :precondition (and
      (at ?from)
      (facing ?dir)
      (move-dir ?from ?to ?dir)
    )
    :effect (and
      (not (at ?from))
      (at ?to)
    )
  )

  (:action open
    :parameters (?pkg - package ?pos - position ?pkgpos - position ?dir -
        direction)
    :precondition (and
      (at ?pos)
      (package-at ?pkg ?pkgpos)
      (package-closed ?pkg)
      (facing ?dir)
      (move-dir ?pos ?pkgpos ?dir)
```

```
    )
    :effect (and
      (not (package-closed ?pkg))
      (package-open ?pkg)
    )
  )

  (:action close
    :parameters (?pkg - package ?pos - position ?pkgpos - position ?dir -
        direction)
    :precondition (and
      (at ?pos)
      (package-at ?pkg ?pkgpos)
      (package-open ?pkg)
      (facing ?dir)
      (move-dir ?pos ?pkgpos ?dir)
    )
    :effect (and
      (not (package-open ?pkg))
      (package-closed ?pkg)
    )
  )
)
```

