# OpenReview forum: "Simulation to Rules: A Dual-VLM Framework for Formal Visual Planning"
_ICLR.cc/2026/Conference — ICLR 2026 Poster_

### Official Review · Reviewer_rVGA · 2025-10-23

**Soundness:** 2
**Presentation:** 2
**Contribution:** 2
**Rating:** 8
**Confidence:** 2

**Summary:**

This paper presents VLMFP, a framework that uses two Vision-Language Models (VLMs) to automatically convert visual planning problems into formal PDDL domain and problem files. SimVLM is fine-tuned for visual understanding and action simulation. GenVLM generates the PDDL files and refines them based on SimVLM's feedback. The method works by iteratively generating PDDL files, checking their consistency against SimVLM's simulations, and updating them to correct errors. Evaluated on grid-world domains, VLMFP successfully generates valid plans for unseen problem instances and visual appearances, removing the need for predefined domain files or constant access to the environment.

**Strengths:**

It autonomously generates both PDDL domain and problem files from visual input, eliminating the need for human-predefined domains—a key bottleneck in prior work.

The separation into a specialized simulator (SimVLM) and a general-purpose generator (GenVLM) effectively leverages the strengths of different models.

The feedback loop that compares PDDL execution with VLM simulation is a reasonable mechanism for catching and correcting errors in the generated artefacts.

The framework demonstrates robust performance across unseen problem instances, visual appearances, and even some modified rules within the tested grid-world environments only.

It provides a preliminary basis for bridging the gap between high-dimensional visual perception and precise, symbolic reasoning.

**Weaknesses:**

The evaluation is confined to synthetic 2D grid-worlds. It remains unproven whether the approach can scale to realistic 3D environments, cluttered scenes, or tasks requiring complex physics reasoning.

The method likely relies on the discrete, structured nature of grid cells and positional predicates (e.g., pos-1-1). Translating continuous, real-world spaces into such a formalism is a major unsolved challenge.

SimVLM was fine-tuned on a massive, domain-specific dataset (430k samples). Curating a similar dataset for every new real-world domain (e.g., kitchen manipulation, navigation) would be prohibitively expensive.

The iterative process of generating, executing, and comparing action sequences for refinement is computationally intensive and may not be feasible for real-time applications.

While it handled some rule variations, performance dropped significantly on complex, novel rules (e.g., the "freezing" mechanic), indicating that reasoning about entirely new dynamics is still a limitation.

**Questions:**

1) The framework is demonstrated on 2D grid-worlds. What are the most significant technical hurdles you foresee in scaling this approach to more complex, realistic 3D environments, such as a simulated kitchen or a robot navigation task?

2) The current method relies on discrete positions (e.g., pos-1-1). How could the approach be adapted to handle continuous state spaces, which are common in real-world robotics and control problems?

3) The current setup assumes a full, top-down view of the entire state. How would VLMFP need to be modified to handle partially observable environments where the agent's view is limited?

4) SimVLM required a massive, fine-tuned dataset (430k samples) for the grid-world domains. Is this level of domain-specific data a fundamental requirement, or do you see a path towards a more general-purpose SimVLM that could be applied to novel domains with minimal fine-tuning?

5) The grid-worlds have a fixed set of object types with clear visual representations. How would the object recognition and spatial reasoning capabilities of SimVLM need to improve to handle novel, real-world objects with diverse and often ambiguous appearances?

6) The iterative refinement process involving multiple VLM calls and PDDL simulations seems computationally expensive. What is the latency of the full VLMFP pipeline, and is it feasible for any real-time decision-making scenarios?

7) Could the "simulation consistency checking" step be made more sample-efficient? Are there smarter strategies for generating the action sequences used for comparison, rather than random sampling, to identify logical flaws faster?

8) The results show a sharp performance drop on the completely unseen "freezing" rule (Rule 5). Does this indicate a fundamental limitation in the system's ability to reason about truly novel action dynamics, as opposed to parametric variations of known rules?

9) How robust is the iterative refinement process to persistent or cascading errors? If SimVLM itself makes a systematic error in simulation, could this lead GenVLM to converge on an incorrect but self-consistent PDDL domain?

10) In your view, what single advancement—whether in model architecture, training data, or the core algorithm—would be the most critical for bridging the gap from these compelling grid-world results to a useful real-world application?

---

> ### Author Response · Authors · 2025-11-23
> **Response to Reviewer rVGA (Summary, Q1)**
>
> We thank the reviewer for the insightful comments and valuable feedback! Your questions and suggestions have greatly strengthened our work.
> We
> * Added an additional 3D assembly task to show that the proposed **VLMFP framework is capable of solving complex task planning in 3D domains** (rVGA-A1) and has great potential to solve even more complex tasks such as under partially observable environments (rVGA-A2)
> * Evaluated SimVLM across different seeds to show that **SimVLM is empirically stable across various seeds, greatly outperforming the baselines** (rVGA-A8)
> * Provided runtime statistics and analysis (rVGA-A5)
> * Provided discussions and analysis about VLMFP’s challenges, potentials, and possible future extensions. (rVGA-A3, rVGA-A4, rVGA-A6, rVGA-A7, rVGA-A9)
>
> Please also see the Revision Summary for additional experiments suggested by other reviewers. We address the reviewer's concerns below. We also updated a revised draft and colored the modifications/additional results and discussions with blue.
>
> **rVGA-Q1: The evaluation is confined to synthetic 2D grid-worlds. It remains unproven whether the approach can scale to realistic 3D environments, cluttered scenes, or tasks requiring complex physics reasoning. The method likely relies on the discrete, structured nature of grid cells and positional predicates (e.g., pos-1-1). Translating continuous, real-world spaces into such a formalism is a major unsolved challenge. What are the most significant technical hurdles you foresee in scaling this approach to more complex, realistic 3D environments, such as a simulated kitchen or a robot navigation task? How could the approach be adapted to handle continuous state spaces, which are common in real-world robotics and control problems?**
>
> rVGA-A1:
> We thank the reviewer for the valuable suggestions. We claim and show with **experiments of a 3D assembly task** that **VLMFP is not restricted to grid-world domains**, and it has the potential to scale to 3D continuous tasks, even with partially observable environments.
> To prove our claim, we evaluate our framework with a **complex long-horizon 3D assembly task** (from Reflective Planning [1]). We updated our draft to include this experiment and kindly refer the reviewer to Section 5.6 and Appendix C.7 for **visualizations** and details of the task.
> * Domain description
>     * In the scenario, there is a puzzle consisting of a board and several blocks with different colors on the table. The goal is to assemble the puzzle to reach the goal configuration with the robot arm. The robot arm can pick up, put down, insert, or reorient the block. The blocks have dependence on other blocks; that is, to achieve the goal, the blocks should be inserted after the blocks they depend on are inserted.
> * Challenges compared to existing grid-world evaluations
>     * **3D continuous realistic task**
>     * **No natural language description for goal configuration.** Since the Reflective Planning paper uses a goal image, instead of a clear natural language description, to represent the goal configuration, which is a much more challenging task, as the VLM would need to recognize, summarize, or even infer the dependency relationships between blocks of goal configuration. We keep this challenging setup to show the flexibility of SimVLM and VLMFP. Thus, SimVLM takes in both initial and goal configuration images as the inputs.
>     * **All are randomized**: block number, color, positions, shape, state; dependence order;  board color, position, shape;
>     * **Occlusion, thus partially observable.** Although not always, sometimes the stacked blocks could occlude each other.
> * Evaluation and analysis
>     * SimVLM: we collected ~25k datapoints and fine-tuned Qwen2-VL-7B as our SimVLM. The appearance differences are created through randomizing color/ position/ order/ number with different seeds. Similar to our previous evaluation, we tested our SimVLM with 1000 datapoints each from seen (S) and unseen (U) appearances.
>     * As shown in the table, SimVLM achieves consistently strong performance across all four evaluation metrics, with an average of 96.0% and 84.8% for seen and unseen appearances. The only visible decline appears in task-description accuracy on unseen appearances, but this is largely an artifact of our strict exact-match evaluation. Thus, even if a single sentence of description is missing, the description would be considered a failure. However, omissions in high-level descriptions do not necessarily affect downstream PDDL generation, since many objects mentioned in the description may be irrelevant to the specific plan. This is reflected in VLMFP’s end-to-end performance: despite the strict scoring on description matching, the overall planning success remains high across both seen and unseen settings.
>     * VLMFP: Similarly to the paper, we evaluate success rates for 1500 trials each for S and U. The success rates are **92.0%** and **85.0%**, respectively.
>
> -rVGA-A1 continues -

---

> > ### Author Response · Authors · 2025-11-23
> > **Response to Reviewer rVGA (Q1 continues, Q2, Q3)**
> >
> > -rVGA-Q2 continues-
> >
> > * Are 2D gridworld scenarios simpler?
> >     * **For PDDL-style task planning, complexity does not grow primarily with spatial dimensionality (2D vs. 3D), but with the richness of the symbolic structure being modeled.** In our framework, complexity comes from
> >         * (1) **scenario understanding and action simulation**, which depend on factors such as the number of objects, object types, spatial relations, etc.; and
> >         * (2) **domain formalization**, which depends on the complexity of action mechanisms, object interactions, rule variations, etc.
> >
> >     These sources of complexity exist regardless of whether the scene is rendered in 2D or 3D. To show the flexibility of our framework, we focus on **diverse domains with heterogeneous mechanisms, action types, and rule structures**, which represents complexity that is ***orthogonal*** to geometric dimensionality.
> > * Potential future extension
> >     * We believe that extending our framework to handle more diverse and realistic 3D tasks is an important direction for future work, particularly given the rich availability of real-world 3D vision and robotics datasets. We demonstrate that SimVLM can process both the initial and goal images as inputs. This suggests that the model can naturally extend to **multi-view inputs for handling occlusions** and to **temporal image sequences** that capture how a scene evolves over time, enabling planning in dynamic environments. We would love to explore these directions further in the future. However, we would like to emphasize that **establishing a reliable and generalizable visual-to-symbolic planning pipeline is itself a significant step for VLM-based long-horizon planning**, and it lays a solid foundation upon which many future extensions can be built.
> >
> > **Table 1: String matching rate (%) of SimVLM and Success rate (%) of VLMFP on Assembly.**
> >
> > | Domain   | Task Description | Execution Reason | Execution Result | Goal Reach | Average | VLMFP |
> > |----------|---------------|--------------|--------------|------------|---------|-------|
> > | Assembly(S)| 86.9          | 99.2         | 99.2         | 98.6       | 96.0    | 92.0  |
> > | Assembly(U)| 52.9          | 94.9         | 95.4         | 96.0       | 84.8    | 85.0  |
> >
> >
> > [1] Feng, Yunhai, et al. "Reflective planning: Vision-language models for multi-stage long-horizon robotic manipulation." Proceedings of the 9th Conference on Robot Learning (CoRL), 2025.
> >
> > **rVGA-Q2: The current setup assumes a full, top-down view of the entire state. How would VLMFP need to be modified to handle partially observable environments where the agent's view is limited?**
> >
> > rVGA-A2: As discussed in rVGA-A1, the Assembly task already exhibits partial observability due to occlusions. We have noted that the model can naturally extend to **multi-view inputs** to further mitigate such occlusions. Beyond this, the agent’s observation may also change during execution, revealing new information over time. This can likewise be handled by providing SimVLM with multiple observation images, where the **combined visual inputs** capture the full underlying state. We would love to explore this further in the future.
> >
> > **rVGA-Q3: SimVLM was fine-tuned on a massive, domain-specific dataset (430k samples). Curating a similar dataset for every new real-world domain (e.g., kitchen manipulation, navigation) would be prohibitively expensive. Is this level of domain-specific data a fundamental requirement, or do you see a path towards a more general-purpose SimVLM that could be applied to novel domains with minimal fine-tuning?**
> >
> > rVGA-A3: We do not view a large in-domain dataset as a fundamental requirement of our framework. In practice, we observed during development that SimVLM’s generalization ability improves substantially as it is exposed to more diverse domains, map structures, and rule systems. This suggests that a more universal perceptual–simulation module is achievable when stronger base VLMs are trained or fine-tuned on increasingly heterogeneous environments.
> >
> > Moreover, SimVLM operates only on a single initial-state image per instance and outputs structured symbolic consequences rather than pixel-level predictions. This makes its data requirement significantly lighter than full world-model learning. In addition, large-scale 3D vision and robotics datasets already contain rich multi-view observations that can support broadly generalizable perceptual grounding.
> >
> > Recent progress in VLA/VLM-based world models further demonstrates that training on diverse environments yields strong cross-domain transfer with minimal fine-tuning. These trends indicate a clear path toward a general-purpose SimVLM that can be applied to novel domains with far less domain-specific data.

---

> > > ### Author Response · Authors · 2025-11-23
> > > **Response to Reviewer rVGA (Q4, Q5, Q6)**
> > >
> > > **rVGA-Q4: The grid-worlds have a fixed set of object types with clear visual representations. How would the object recognition and spatial reasoning capabilities of SimVLM need to improve to handle novel, real-world objects with diverse and often ambiguous appearances?**
> > >
> > > rVGA-A4:
> > > Although grid-worlds use a fixed set of object types, our current SimVLM shows how it generalizes well to unseen appearances with new colors, textures, or even new visual styles (‘Player’ appears as a human and robot dog in different appearances), showing that it can handle variations in object look without domain-specific tuning. Extending this capability to real-world objects primarily requires exposing the model to broader visual diversity.
> > >
> > > One practical extension is to provide **object reference images** directly in the prompt. This matching of object type and its look can work as clear object descriptions that guide SimVLM. Many real-world objects can be described using cropped views or representative images. This matching of object type and its look can work as a guidance to SimVLM.
> > >
> > > Further improvements , such as stronger visual encoders and training on diverse 3D vision or robotics datasets, are consistent with trends in large embodied VLMs, which already demonstrate cross-domain generalization with minimal fine-tuning. Thus, we see a clear path toward scaling SimVLM to real-world object diversity.
> > >
> > > **rVGA-Q5: The iterative process of generating, executing, and comparing action sequences for refinement is computationally intensive and may not be feasible for real-time applications. What is the latency of the full VLMFP pipeline, and is it feasible for any real-time decision-making scenarios?**
> > >
> > > rVGA-A5:
> > > We report the average latency of VLM calls (SimVLM and GenVLM) in the full pipeline for the 15 input instances in the following table. Since the most expensive components are the VLM forward passes, these timings exclude overhead such as model loading, checkpoint initialization, and general program overhead.
> > >
> > > **Table 2: Latency (s) of VLM calls of VLMFP across 6 domains**
> > >
> > > | VLM Call runtime   | Frozenlake | Maze | Sokoban | Package | Printer | Overcooked | **Average** |
> > > |----------|---------------|--------------|--------------|------------|---------|-------|-------|
> > > |Latency(S)|203.0|223.2|353.3|370.2|360.9|286.1|**299.5**|
> > > |Latency(U)|188.6|224.0|364.9|335.6|349.3|363.8|**304.2**|
> > >
> > > On average, VLMFP takes around 300 seconds to generate domain and problem PDDL files for input instances. However, we would like to emphasize that **once the domain PDDL file is constructed, solving a new instance within the same domain requires only generating the problem file**, which is a single SimVLM pass and therefore dramatically faster.
> > >
> > > In other words, the iterative refinement loop is a one-time offline cost for each domain. VLMFP is intended for offline model construction rather than real-time control. After this initial phase, the framework supports **fast, reusable planning across many instances without further iterative simulation or refinement**.
> > >
> > > **rVGA-Q6: While it handled some rule variations, performance dropped significantly on complex, novel rules (e.g., the "freezing" mechanic), indicating that reasoning about entirely new dynamics is still a limitation. The results show a sharp performance drop on the completely unseen "freezing" rule (Rule 5). Does this indicate a fundamental limitation in the system's ability to reason about truly novel action dynamics, as opposed to parametric variations of known rules?**
> > >
> > > rVGA-A6:
> > > We agree that performance drops on the "freezing" rule reveal important boundaries of the current approach. However, the failure mode is revealing: SimVLM achieves 71.1% reasoning accuracy (correctly stating "next action is skipped") but 0% execution success. This indicates SimVLM **understands the rule linguistically** but cannot integrate it into a multi-step simulation. We believe this is a common and expected limitation of learned simulators on out-of-distribution dynamics rather than a structural limitation of VLMFP. Rules 1-4 involve **recombining familiar concepts** (movement, teleportation, repetition) in new ways. Rule 5 introduces a **fundamentally new state variable** ('frozen status') with temporal dependencies never observed in training. This distinction is critical.
> > >
> > > Increasing the diversity of training domains, incorporating in-context examples by including examples of the new rule, and incorporating meta-learning would have great potential to allow the framework to acquire the corresponding causal pattern. We would love to explore these directions in the future. However, we believe that **establishing a reliable and generalizable visual-to-symbolic planning pipeline is itself a significant step for VLM-based long-horizon planning**, and it lays a solid foundation upon which many future extensions can be built.

---

> > > > ### Author Response · Authors · 2025-11-23
> > > > **Response to Reviewer rVGA (Q7, Q9)**
> > > >
> > > > **rVGA-Q7: Could the "simulation consistency checking" step be made more sample-efficient? Are there smarter strategies for generating the action sequences used for comparison, rather than random sampling, to identify logical flaws faster?**
> > > >
> > > > rVGA-A7:
> > > > We agree that there is room for more sample-efficient strategies. For example, we see several straightforward ways to make the consistency checking more targeted:
> > > > * **Coverage-based action selection.** Rather than sampling uniformly, one can maintain statistics over which actions, objects, and predicates have been exercised and prioritize sequences that involve under-tested components (rarely used actions or object types). This would improve logical coverage without substantially increasing the number of rollouts.
> > > > * **LLM-guided boundary tests.** One could use an LLM to propose action sequences that are likely to expose rule boundaries (e.g., moves near walls), explore actions with complex mechanisms, or toward predicates that are often involved in failures.
> > > >
> > > > We view these as natural extensions of VLMFP’s consistency-checking step, which is a promising direction for future improvements.
> > > >
> > > > **rVGA-Q9: In your view, what single advancement—whether in model architecture, training data, or the core algorithm—would be the most critical for bridging the gap from these compelling grid-world results to a useful real-world application?**
> > > >
> > > > rVGA-A9:
> > > > From my perspective, the most important step is making SimVLM a more general and perception-robust world model by training it on a wider range of real-world visual data. If SimVLM can reliably interpret scenes and anticipate action outcomes across diverse environments, then the rest of the VLMFP framework would transfer much more naturally from grid worlds to real applications.

---

> > > > > ### Author Response · Authors · 2025-11-23
> > > > > **Response to Reviewer rVGA (Q8)**
> > > > >
> > > > > **rVGA-Q8: How robust is the iterative refinement process to persistent or cascading errors? If SimVLM itself makes a systematic error in simulation, could this lead GenVLM to converge on an incorrect but self-consistent PDDL domain?**
> > > > >
> > > > > rVGA-A8:
> > > > > We conduct additional experiments to evaluate SimVLM across **3 random seeds** to measure its stability. As shown in the table below, across the 6 domains, the standard deviation for all four metrics is very small, indicating that **SimVLM’s perception, step-wise reasoning, and action-result predictions are highly stable and SimVLM does not hallucinate unpredictably**. We have added this multi-seed experiment result in Appendix C.3.
> > > > >
> > > > > **Table 3: The mean of string matching rate (%) for 4 SimVLM output types on 6 grid-world domains across 3 seeds.**
> > > > >
> > > > > | Output           | Frozenlake S | Frozenlake U | Maze S | Maze U | Sokoban S | Sokoban U | Package S | Package U | Printer S | Printer U | Overcooked S | Overcooked U | Avg S | Avg U |
> > > > > |------------------|--------------|--------------|--------|--------|-----------|-----------|-----------|-----------|-----------|-----------|--------------|--------------|--------|--------|
> > > > > | Task Descrip.    | 99.9         | 91.9         | 99.2   | 89.9   | 74.0      | 51.4      | 100.0     | 99.9      | 99.8      | 96.6      | 99.2         | 64.9         | **95.3**   | **82.4**   |
> > > > > | Exec. Reason     | 94.9         | 94.8         | 83.6   | 96.9   | 72.4      | 83.4      | 88.5      | 89.0      | 85.7      | 85.4      | 76.9         | 75.8         | **83.7**   | **87.5**   |
> > > > > | Exec. Result     | 94.9         | 94.7         | 83.6   | 97.0   | 72.1      | 83.0      | 88.5      | 89.0      | 85.7      | 85.4      | 77.0         | 75.9         | **83.7**   | **87.5**   |
> > > > > | Goal Reach       | 93.5         | 93.5         | 80.3   | 94.7   | 72.7      | 83.1      | 86.8      | 86.6      | 83.6      | 83.7      | 73.9         | 73.2         | **81.8**   | **85.8**   |
> > > > > | **Average**      | **95.8**     | **93.7**     | **86.7** | **94.6** | **72.8** | **75.2** | **91.0** | **91.1** | **88.7** | **87.8** | **81.8**     | **72.4**     | **86.1** | **85.8** |
> > > > >
> > > > >
> > > > > **Table 4: The standard deviation of string matching rate (%) for 4 SimVLM output types on 6 grid-world domains across 3 seeds.**
> > > > >
> > > > > | Output           | Frozenlake S | Frozenlake U | Maze S | Maze U | Sokoban S | Sokoban U | Package S | Package U | Printer S | Printer U | Overcooked S | Overcooked U | Avg S | Avg U |
> > > > > |------------------|--------------|--------------|--------|--------|-----------|-----------|-----------|-----------|-----------|-----------|--------------|--------------|--------|--------|
> > > > > | Task Descrip.    | 0.1          | 0.5          | 0.1    | 0.1    | 2.1       | 0.3       | 0.0       | 0.2       | 0.1       | 0.4       | 0.2          | 1.1          | **0.3**    | **0.2**    |
> > > > > | Exec. Reason     | 1.5          | 1.3          | 1.5    | 0.2    | 3.8       | 2.0       | 2.3       | 1.0       | 2.2       | 2.0       | 1.8          | 3.7          | **2.0**    | **1.3**    |
> > > > > | Exec. Result     | 1.4          | 1.3          | 1.5    | 0.2    | 3.6       | 0.6       | 2.3       | 1.0       | 2.2       | 2.0       | 1.6          | 3.7          | **2.0**    | **1.0**    |
> > > > > | Goal Reach       | 0.5          | 0.2          | 1.7    | 0.0    | 1.4       | 0.1       | 1.1       | 0.7       | 0.5       | 0.8       | 1.9          | 2.4          | **0.7**    | **0.2**    |
> > > > > | **Average**      | **0.8**      | **0.7**      | **0.7** | **0.0** | **2.2**   | **0.6**   | **1.4**   | **0.7**   | **1.2**   | **1.1**   | **0.9**      | **2.5**      | **1.1** | **0.6** |
> > > > >
> > > > > Additionally, we emphasize that the SimVLM metrics are computed over **1000 evaluation datapoints**, and VLMFP results are averaged over **1500 planning trials** for each domain. These large-sample evaluations already demonstrate strong empirical stability despite the possibility of systematic error raised by the reviewer.
> > > > >
> > > > > Finally, as discussed in Section X.X, the iterative refinement loop is not solely driven by execution mismatches. In each iteration, GenVLM is explicitly prompted to systematically inspect all critical components by listing and reasoning whether the objects, object types, init states, goal states, predicates, and actions are valid and coherent, identifies which file(s) are erroneous, and applies targeted modifications. This structured inspection helps identify inconsistencies even when the execution feedback signal is weak or noisy. As a result, the refinement process does not blindly propagate SimVLM errors. Instead, it leverages both execution-level discrepancies and higher-level logical checks to localize and correct mistakes before updating the PDDL files. This makes it less vulnerable to erroneous SimVLM simulation.

---

> > > > > > ### Comment · Reviewer_rVGA · 2025-11-27
> > > > > >
> > > > > > Dear authors, Thanks for the clarifications. I will consider them for revising my review. Best

---

### Official Review · Reviewer_puu1 · 2025-10-26

**Soundness:** 3
**Presentation:** 1
**Contribution:** 3
**Rating:** 2
**Confidence:** 4

**Summary:**

This paper investigates translating 2D navigation planning environments into formal symbolic PDDL representations. The task is meaningful because accurate translation enables invoking PDDL solvers for automated planning. The proposed approach uses an iterative refinement process: a VLM first generates draft PDDL files, then executes random actions in the environment to verify whether the outcomes align with simulation results. Experiments show that the method achieves a reasonable success rate (70% on average).

Overall, the experimental results are generally sufficient, but the writing is poor and difficult to follow. The paper requires a major revision to improve clarity and readability before it can be considered for publication.

**Strengths:**

- This is an interesting and novel research direction, to the best of my knowledge.
- Based on the experimental results, the proposed method appears to significantly improve translation accuracy compared to existing approaches.

**Weaknesses:**

- The writing can be improved. The paper is dense and difficult to follow, focusing heavily on implementation details while lacking a clear, high-level conceptual framework. It is hard to identify the core scientific research question amid the technical descriptions. The method section is overly detailed, and the ideas could be presented more clearly by abstracting the core principles. For example, the iterative refinement process could be framed as learning a formal world model (domain files) and structured state representations (problem files). A clearer separation between methodology and implementation would significantly improve readability.

- It is unclear whether dividing the process into two VLMs, SimVLM and GenVLM, is necessary. In principle, a single large VLM with self-refinement capabilities could fulfill both roles. Using a smaller model for specific subtasks makes sense from a computational efficiency perspective, but this seems more like an engineering optimization than a conceptual requirement. If the authors wish to emphasize cost-effectiveness as a contribution, this should be highlighted in the abstract and introduction, and supported with experiments comparing computation cost and accuracy between small and large VLMs.

- I recommend adding a discussion on extending this framework to more complex and realistic domains, such as robotic control. A common criticism of PDDL-based approaches is their limited flexibility; this framework could address that issue. It would strengthen the paper to explicitly discuss how VLMFP might generalize beyond grid-world settings.

- Some suggested related works:
    - *Efficient Exploration and Discriminative World Model Learning with an Object-Centric Abstraction*
    - *Planning with Reasoning using Vision-Language World Model*
    - *Predicate Invention for Bilevel Planning*

**Questions:**

See Weaknesses

---

> ### Author Response · Authors · 2025-11-23
> **Response to Reviewer puu1 (Summary and Q1)**
>
> We thank the reviewer for the valuable comments and suggestions!
>
> We
> * Added an additional 3D assembly task to show that the proposed **VLMFP framework is capable of solving complex task planning in 3D domains** (puu1-A3)
> * Revised the paper to provide a clear high-level conceptual framework overview and separate method with implementation details (puu1-A1); added the related works mentioned by the reviewer (puu1-A4)
> * Provide some clarifications and discussions of the dual-VLM system (puu1-A2)
>
> Please also see the Revision Summary for additional experiments suggested by other reviewers. We address the reviewer's concerns below. We also updated a revised draft and colored the modifications/additional results and discussions with blue.
>
> **puu1-Q1: The writing can be improved.**
>
> puu1-A1:
> Thanks for the suggestion. We have modified the method section. In particular,
> * We added a high-level conceptual overview of VLMFP to provide an abstract, conceptual, and clean narrative of VLMFP and its objective.
> * We modified the narratives within subsections of the method section to make it more logical and smooth.
> * We separated the implementation details to be another section.
>
> Please refer to our updated draft. The changes are highlighted in blue.

---

> ### Author Response · Authors · 2025-11-23
> **Response to Reviewer puu1 (Q2)**
>
> **puu1-Q2: It is unclear whether dividing the process into two VLMs, SimVLM and GenVLM, is necessary.**
>
> puu1-A2:
> The reasons why dividing the process into two VLMs (small+large) is necessary are:
> * The two VLMs have **fundamentally different roles** and **conflicting comparative advantages**,
> * **Large VLMs cannot replace SimVLM**: We empirically show that general API-based large VLMs **lack precise spatial understanding and long-horizon reasoning capabilities** to directly accomplish the task of SimVLM, thus fine-tuning a specialized model is necessary.
> * **Small fine-tuned VLMs cannot replace GenVLM**: Large VLM’s **PDDL knowledge** and **general reasoning capabilities** are necessary for GenVLM.
>
> We explain in detail with experiments as follows:
> * As we described in line 220-223, SimVLM should be stronger at **precisely understanding the spatial relationship in images**, and have a superior capability in **simulating action consequences**. GenVLM should possess **stronger general reasoning and question-answering capabilities**, and have **richer knowledge in PDDL**.
> * We include the comparison with two more baselines: 1) Direct_GPT-4o: using GPT4o to directly accomplish the task, given the expected format; and 2) InContext_GPT-4o: using GPT4o to accomplish the task, given the expected format and an in-context example. Note that we use the same four metrics (Task Description, Execution Reason, Execution Result, Goal Reach) for comparison. However, since for Execution Reason, we do not expect API-based VLMs to output exact formatted output as the ground truth output, we did not use exact string match. Instead, we use the sentence-transformers library with all-MiniLM-L6-v2 model to check the Semantic Similarity for Execution Reason, which is a less strict evaluation than SimVLM. From the result, Direct_GPT-4o and InContext_GPT-4o can only accomplish the four tasks with an average of 2.5% and 3.3%(extremely low because this is not binary judgement), 51.6% and 76.5%(semantic similarity), 35.7% and 67.3%(binary judgement), 81.4% and 87.7%(binary judgement), comparing to 82.6%, 88.1%,  87.8%,  85.6% of SimVLM. From the results, we can observe that general VLMs still **lack precise spatial understanding and long-horizon reasoning capabilities**. While after observing input and output pairs of a wide range of different domains with various appearances, a small fine-tuned VLM learned how to map from **images → structured spatial representations → action-conditioned transitions**, which is not taught during web-scale training. In our paper, we empirically show that it is stable and generalizable. Thus, large VLMs cannot replace SimVLM.
>
>
> **Table 1: String matching rate (%) comparison of SimVLM and baselines across four metrics on 6 grid-world domains.**
>
> ### **Task Description**
> | Method | Frozenlake | Maze | Sokoban | Package | Printer | Overcooked | Average |
> |--------|------------|------|---------|---------|---------|------------|---------|
> | Direct_GPT-4o | 0.0 | 0.0 | 0.0 | 12.7 | 2.0 | 0.0 | 2.5 |
> | InContext_GPT-4o | 0.0 | 0.0 | 0.0 | 12.0 | 8.0 | 0.0 | 3.3 |
> | **SimVLM** | 92.3 | 89.8 | 51.6 | 99.7 | 96.3 | 65.6 | 82.6 |
>
> ---
>
> ### **Execution Reason**
> | Method | Frozenlake | Maze | Sokoban | Package | Printer | Overcooked | Average |
> |--------|------------|------|---------|---------|---------|------------|---------|
> | Direct_GPT-4o | 34.6 | 63.4 | 21.7 | 55.6 | 63.4 | 70.7 | 51.6 |
> | InContext_GPT-4o | 47.8 | 82.7 | 82.9 | 79.5 | 82.4 | 83.8 | 76.5 |
> | **SimVLM** | 96.0 | 97.1 | 84.6 | 88.4 | 86.8 | 75.9 | 88.1 |
>
> ---
>
> ### **Execution Result**
> | Method | Frozenlake | Maze | Sokoban | Package | Printer | Overcooked | Average |
> |--------|------------|------|---------|---------|---------|------------|---------|
> | Direct_GPT-4o | 29.1 | 42.9 | 16.9 | 23.9 | 56.6 | 44.8 | 35.7 |
> | InContext_GPT-4o | 32.1 | 63.0 | 73.4 | 82.8 | 83.6 | 68.9 | 67.3 |
> | **SimVLM** | 95.9 | 96.8 | 83.4 | 88.4 | 86.8 | 75.7 | 87.8 |
>
> ---
>
> ### **Goal Reach**
> | Method | Frozenlake | Maze | Sokoban | Package | Printer | Overcooked | Average |
> |--------|------------|------|---------|---------|---------|------------|---------|
> | Direct_GPT-4o | 72.0 | 68.8 | 80.8 | 79.2 | 91.4 | 96.0 | 81.4 |
> | InContext_GPT-4o | 83.8 | 74.4 | 96.8 | 83.8 | 89.0 | 99.2 | 87.7 |
> | **SimVLM** | 93.7 | 94.7 | 83.2 | 86.0 | 84.6 | 71.2 | 85.6 |
>
> * Similarly, as the small VLM is fine-tuned specifically for spatial recognition and reasoning, its general reasoning capability and PDDL knowledge is prone to catastrophic forgetting. Thus, Small fine-tuned VLM also cannot replace GenVLM.
>
> To summarize, splitting into two VLMs is necessary as they have different and conflicting advantages, and the small VLM and large VLM cannot replace each other.

---

> > ### Author Response · Authors · 2025-11-23
> > **Response to Reviewer puu1 (Q3)**
> >
> > **puu1-Q3: I recommend adding a discussion on extending this framework to more complex and realistic domains, such as robotic control.**
> >
> > puu1-A3:
> > We thank the reviewer for the valuable suggestions. We claim and show with **experiments of a 3D assembly task** that **VLMFP is not restricted to grid-world domains**, and it has the potential to scale to 3D continuous tasks, even with partially observable environments.
> > To prove our claim, we evaluate our framework with a **complex long-horizon 3D assembly task** (from Reflective Planning [1]). We updated our draft to include this experiment and kindly refer the reviewer to Section 5.6 and Appendix C.7 for **visualizations** and details of the task.
> > * Domain description
> >     * In the scenario, there is a puzzle consisting of a board and several blocks with different colors on the table. The goal is to assemble the puzzle to reach the goal configuration with the robot arm. The robot arm can pick up, put down, insert, or reorient the block. The blocks have dependence on other blocks; that is, to achieve the goal, the blocks should be inserted after the blocks they depend on are inserted.
> > * Challenges compared to existing grid-world evaluations
> >     * **3D continuous realistic task**
> >     * **No natural language description for goal configuration.** Since the Reflective Planning paper uses a goal image, instead of a clear natural language description, to represent the goal configuration, which is a much more challenging task, as the VLM would need to recognize, summarize, or even infer the dependency relationships between blocks of goal configuration. We keep this challenging setup to show the flexibility of SimVLM and VLMFP. Thus, SimVLM takes in both initial and goal configuration images as the inputs.
> >     * **All are randomized**: block number, color, positions, shape, state; dependence order;  board color, position, shape;
> >     * **Occlusion, thus partially observable.** Although not always, sometimes the stacked blocks could occlude each other.
> > * Evaluation and analysis
> >     * SimVLM: we collected ~25k datapoints and fine-tuned Qwen2-VL-7B as our SimVLM. The appearance differences are created through randomizing color/ position/ order/ number with different seeds. Similar to our previous evaluation, we tested our SimVLM with 1000 datapoints each from seen (S) and unseen (U) appearances.
> >     * As shown in the table, SimVLM achieves consistently strong performance across all four evaluation metrics, with an average of 96.0% and 84.8% for seen and unseen appearances. The only visible decline appears in task-description accuracy on unseen appearances, but this is largely an artifact of our strict exact-match evaluation. Thus, even if a single sentence of description is missing, the description would be considered a failure. However, omissions in high-level descriptions do not necessarily affect downstream PDDL generation, since many objects mentioned in the description may be irrelevant to the specific plan. This is reflected in VLMFP’s end-to-end performance: despite the strict scoring on description matching, the overall planning success remains high across both seen and unseen settings.
> >     * VLMFP: Similarly to the paper, we evaluate success rates for 1500 trials each for S and U. The success rates are **92.0%** and **85.0%**, respectively.
> >
> > **Table 2: String matching rate (%) of SimVLM and Success rate (%) of VLMFP on Assembly.**
> >
> > | Domain   | Task Description | Execution Reason | Execution Result | Goal Reach | Average | VLMFP |
> > |----------|---------------|--------------|--------------|------------|---------|-------|
> > | Assembly (S)| 86.9          | 99.2         | 99.2         | 98.6       | 96.0    | 92.0  |
> > | Assembly (U)| 52.9          | 94.9         | 95.4         | 96.0       | 84.8    | 85.0  |
> >
> > puu1-A3 continues next page

---

> > > ### Author Response · Authors · 2025-11-23
> > > **Response to Reviewer puu1 (Q3 continue, Q4)**
> > >
> > > puu1-A3 continues
> > >
> > > * Are 2D gridworld scenarios simpler?
> > >     * **For PDDL-style task planning, complexity does not grow primarily with spatial dimensionality (2D vs. 3D), but with the richness of the symbolic structure being modeled.** In our framework, complexity comes from
> > >         * (1) **scenario understanding and action simulation**, which depend on factors such as the number of objects, object types, spatial relations, etc.; and
> > >         * (2) **domain formalization**, which depends on the complexity of action mechanisms, object interactions, rule variations, etc.
> > >
> > >     These sources of complexity exist regardless of whether the scene is rendered in 2D or 3D. To show the flexibility of our framework, we focus on **diverse domains with heterogeneous mechanisms, action types, and rule structures**, which represents complexity that is ***orthogonal*** to geometric dimensionality.
> > > * Potential future extension
> > >     * We believe that extending our framework to handle more diverse and realistic 3D tasks is an important direction for future work, particularly given the rich availability of real-world 3D vision and robotics datasets. We demonstrate that SimVLM can process both the initial and goal images as inputs. This suggests that the model can naturally extend to **multi-view inputs for handling occlusions** and to **temporal image sequences** that capture how a scene evolves over time, enabling planning in dynamic environments. We would love to explore these directions further in the future. However, we would like to emphasize that **establishing a reliable and generalizable visual-to-symbolic planning pipeline is itself a significant step for VLM-based long-horizon planning**, and it lays a solid foundation upon which many future extensions can be built.
> > >
> > > [1] Feng, Yunhai, et al. "Reflective planning: Vision-language models for multi-stage long-horizon robotic manipulation." Proceedings of the 9th Conference on Robot Learning (CoRL), 2025.
> > >
> > > **puu1-Q4: Some suggested related works**
> > >
> > > puu1-A4: Thank you for your suggestion. We have added the mentioned related works in Section 2 of our updated draft.

---

### Official Review · Reviewer_8x3c · 2025-10-31

**Soundness:** 3
**Presentation:** 3
**Contribution:** 2
**Rating:** 4
**Confidence:** 3

**Summary:**

This paper introduces VLM-Guided Formal Planning (VLMFP), a novel framework for the autonomous generation of PDDL files from visual inputs.

Existing VLM-based approaches in visual planning are limited in their ability to accurately generate domain PDDL and often rely on access to ground-truth domain files.
To overcome these limitations, VLMFP employs a dual-VLM architecture: a fine-tuned SimVLM that perceives environmental scenarios from visual inputs and simulates action outcomes, alongside a large generative VLM (GenVLM) that generates and iteratively refines PDDL files by aligning their execution with SimVLM’s simulations.

Evaluated across six grid world domains, VLMFP demonstrates strong generalization capabilities to unseen instances, visual appearances, and game rules.

**Strengths:**

1. The proposed dual-VLM framework is well-justified.
It effectively allocates distinct reasoning roles to each model: SimVLM, a fine-tuned smaller VLM, specializes in spatial reasoning from visual inputs, while GenVLM, the larger model, leverages SimVLM’s output to reason over and generate PDDL content.

2. The paper is clearly structured.
The introduction and Section 3.1 offer a coherent presentation of the problem formulation, related work, and the motivation behind the proposed approach.

**Weaknesses:**

The paper addresses a highly specific problem, which may limit the broader applicability of the proposed approach.
The evaluation, confined to six grid-world domains, lacks demonstration on more diverse PDDL scenarios.
Furthermore, the requirement for in-domain training data for SimVLM in each new domain could hinder the framework's practical scalability.

**Questions:**

My main questions regarding the evaluation are as follows:

1. In Table 1, the proposed SimVLM component is not compared against any baseline methods.

2. In Table 2, the CodePDDL baseline shows comparable performance to VLMFP on the Frozenlake and Maze domains but performs poorly on others. What is the underlying reason for this significant performance disparity across domains? An explanation is needed to interpret these results correctly.

3. Several relevant papers mentioned in the text (Line 52-79) are not included as baselines in Table 1 or 2. Including these comparisons is crucial for a fair and thorough evaluation of VLMFP's performance against contemporary works.

---

> ### Author Response · Authors · 2025-11-23
> **Response to Reviewer 8x3c (Summary and Q1)**
>
> We thank the reviewer for the constructive comments and helpful feedback! You brought up some great questions and suggestions, which have helped improve our work.
>
> We
> * Added an additional 3D assembly task to show that the proposed **VLMFP framework is capable of solving complex task planning in 3D domains** (8x3c-A1)
> * Conducted baseline comparisons for SimVLM (8x3c-A2)
> * Added additional explanations and analysis for baselines and domain complexity (8x3c-A3, 8x3c-A4)
>
> Please also see the Revision Summary for additional experiments suggested by other reviewers. We address the reviewer's concerns below. We also updated a revised draft and colored the modifications/additional results and discussions with blue.
>
> **8x3c-Q1: The paper addresses a highly specific problem, which may limit the broader applicability of the proposed approach. The evaluation, confined to six grid-world domains, lacks demonstration on more diverse PDDL scenarios. Furthermore, the requirement for in-domain training data for SimVLM in each new domain could hinder the framework's practical scalability.**
>
> 8x3c-A1:
> We thank the reviewer for the valuable suggestions. We claim and show with **experiments of a 3D assembly task** that **VLMFP is not restricted to grid-world domains**, and it has the potential to scale to 3D continuous tasks, even with partially observable environments.
> To prove our claim, we evaluate our framework with a **complex long-horizon 3D assembly task** (from Reflective Planning [1]). We updated our draft to include this experiment and kindly refer the reviewer to Section 5.6 and Appendix C.7 for **visualizations** and details of the task.
> * Domain description
>     * In the scenario, there is a puzzle consisting of a board and several blocks with different colors on the table. The goal is to assemble the puzzle to reach the goal configuration with the robot arm. The robot arm can pick up, put down, insert, or reorient the block. The blocks have dependence on other blocks; that is, to achieve the goal, the blocks should be inserted after the blocks they depend on are inserted.
> * Challenges compared to existing grid-world evaluations
>     * **3D continuous realistic task**
>     * **No natural language description for goal configuration.** Since the Reflective Planning paper uses a goal image, instead of a clear natural language description, to represent the goal configuration, which is a much more challenging task, as the VLM would need to recognize, summarize, or even infer the dependency relationships between blocks of goal configuration. We keep this challenging setup to show the flexibility of SimVLM and VLMFP. Thus, SimVLM takes in both initial and goal configuration images as the inputs.
>     * **All are randomized**: block number, color, positions, shape, state; dependence order;  board color, position, shape;
>     * **Occlusion, thus partially observable.** Although not always, sometimes the stacked blocks could occlude each other.
> * Evaluation and analysis
>     * SimVLM: we collected ~25k datapoints and fine-tuned Qwen2-VL-7B as our SimVLM. The appearance differences are created through randomizing color/ position/ order/ number with different seeds. Similar to our previous evaluation, we tested our SimVLM with 1000 datapoints each from seen (S) and unseen (U) appearances.
>     * As shown in the table, SimVLM achieves consistently strong performance across all four evaluation metrics, with an average of 96.0% and 84.8% for seen and unseen appearances. The only visible decline appears in task-description accuracy on unseen appearances, but this is largely an artifact of our strict exact-match evaluation. Thus, even if a single sentence of description is missing, the description would be considered a failure. However, omissions in high-level descriptions do not necessarily affect downstream PDDL generation, since many objects mentioned in the description may be irrelevant to the specific plan. This is reflected in VLMFP’s end-to-end performance: despite the strict scoring on description matching, the overall planning success remains high across both seen and unseen settings.
>     * VLMFP: Similarly to the paper, we evaluate success rates for 1500 trials each for S and U. The success rates are **92.0%** and **85.0%**, respectively.
>
> **Table 1: String matching rate (%) of SimVLM and Success rate (%) of VLMFP on Assembly.**
>
> | Domain   | Task Description | Execution Reason | Execution Result | Goal Reach | Average | VLMFP |
> |----------|---------------|--------------|--------------|------------|---------|-------|
> | Assembly (S)| 86.9          | 99.2         | 99.2         | 98.6       | 96.0    | 92.0  |
> | Assembly (U)| 52.9          | 94.9         | 95.4         | 96.0       | 84.8    | 85.0  |

---

> > ### Author Response · Authors · 2025-11-23
> > **Response to Reviewer 8x3c (Q1 continue)**
> >
> > 8x3c-A1 - continue:
> >
> > * Are 2D gridworld scenarios simpler?
> >     * **For PDDL-style task planning, complexity does not grow primarily with spatial dimensionality (2D vs. 3D), but with the richness of the symbolic structure being modeled.** In our framework, complexity comes from
> >         * (1) **scenario understanding and action simulation**, which depend on factors such as the number of objects, object types, spatial relations, etc.; and
> >         * (2) **domain formalization**, which depends on the complexity of action mechanisms, object interactions, rule variations, etc.
> >
> >         These sources of complexity exist regardless of whether the scene is rendered in 2D or 3D. To show the flexibility of our framework, we focus on **diverse domains with heterogeneous mechanisms, action types, and rule structures**, which represents complexity that is ***orthogonal*** to geometric dimensionality.
> > * Potential future extension
> >     * We believe that extending our framework to handle more diverse and realistic 3D tasks is an important direction for future work, particularly given the rich availability of real-world 3D vision and robotics datasets. We demonstrate that SimVLM can process both the initial and goal images as inputs. This suggests that the model can naturally extend to **multi-view inputs for handling occlusions** and to **temporal image sequences** that capture how a scene evolves over time, enabling planning in dynamic environments. We would love to explore these directions further in the future.
> >     * Additionally, in the paper, we showed the generalization capabilities of our framework to unseen instances, appearances, and rules. We observe during the development of VLMFP that the generalization capability of SimVLM improved as it saw more diverse domains, map sizes, appearances, and rules. Thus, there is great potential to develop a more generalized simulation ‘world model’ when it exposes stronger VLM base models to increasingly diverse environments.
> >     * However, we would like to emphasize that **establishing a reliable and generalizable visual-to-symbolic planning pipeline is itself a significant step for VLM-based long-horizon planning**, and it lays a solid foundation upon which many future extensions can be built.
> >
> >
> > [1] Feng, Yunhai, et al. "Reflective planning: Vision-language models for multi-stage long-horizon robotic manipulation." Proceedings of the 9th Conference on Robot Learning (CoRL), 2025.

---

> ### Author Response · Authors · 2025-11-23
> **Response to Reviewer 8x3c (Q2)**
>
> **8x3c-Q2: In Table 1, the proposed SimVLM component is not compared against any baseline methods.**
>
> 8x3c-A2: We include the comparison with two more baselines: 1) **Direct_GPT-4o**: using GPT4o to directly accomplish the task, given the expected format; and 2) **InContext_GPT-4o**: using GPT4o to accomplish the task, given the expected format and an in-context example. Note that we use the same four metrics (Task Description, Execution Reason, Execution Result, Goal Reach) for comparison. However, since for Execution Reason, we do not expect API-based VLMs to output exactly formatted output as the ground truth output, we did not use the exact string match. Instead, we use sentence-transformers library with all-MiniLM-L6-v2 model to check the Semantic Similarity for Execution Reason, which is a less strict evaluation than SimVLM. From the result, **Direct_GPT-4o** and **InContext_GPT-4o** can only accomplish the four tasks with an average of 2.5% and 3.3%(extremely low because this is not binary judgement), 51.6% and 76.5%(semantic similarity), 35.7% and 67.3%(binary judgement), 81.4% and 87.7%(binary judgement), comparing to 82.6%, 88.1%,  87.8%,  85.6% of SimVLM. We can observe that the API-based large VLM **lacks precise spatial understanding and long-horizon reasoning capabilities** to directly accomplish the task of SimVLM. Thus, fine-tuning a specialized model is necessary. We also added this additional experiment to Appendix C.1.
>
> **Table 2: String matching rate (%) comparison of SimVLM and baselines across four metrics on 6 grid-world domains.**
>
> ### **Task Description**
> | Method | Frozenlake | Maze | Sokoban | Package | Printer | Overcooked | Average |
> |--------|------------|------|---------|---------|---------|------------|---------|
> | **Direct_GPT-4o** | 0.0 | 0.0 | 0.0 | 12.7 | 2.0 | 0.0 | 2.5 |
> | **InContext_GPT-4o** | 0.0 | 0.0 | 0.0 | 12.0 | 8.0 | 0.0 | 3.3 |
> | **SimVLM** | 92.3 | 89.8 | 51.6 | 99.7 | 96.3 | 65.6 | 82.6 |
>
> ---
>
> ### **Execution Reason**
> | Method | Frozenlake | Maze | Sokoban | Package | Printer | Overcooked | Average |
> |--------|------------|------|---------|---------|---------|------------|---------|
> | **Direct_GPT-4o** | 34.6 | 63.4 | 21.7 | 55.6 | 63.4 | 70.7 | 51.6 |
> | **InContext_GPT-4o** | 47.8 | 82.7 | 82.9 | 79.5 | 82.4 | 83.8 | 76.5 |
> | **SimVLM** | 96.0 | 97.1 | 84.6 | 88.4 | 86.8 | 75.9 | 88.1 |
>
> ---
>
> ### **Execution Result**
> | Method | Frozenlake | Maze | Sokoban | Package | Printer | Overcooked | Average |
> |--------|------------|------|---------|---------|---------|------------|---------|
> | **Direct_GPT-4o** | 29.1 | 42.9 | 16.9 | 23.9 | 56.6 | 44.8 | 35.7 |
> | **InContext_GPT-4o** | 32.1 | 63.0 | 73.4 | 82.8 | 83.6 | 68.9 | 67.3 |
> | **SimVLM** | 95.9 | 96.8 | 83.4 | 88.4 | 86.8 | 75.7 | 87.8 |
>
> ---
>
> ### **Goal Reach**
> | Method | Frozenlake | Maze | Sokoban | Package | Printer | Overcooked | Average |
> |--------|------------|------|---------|---------|---------|------------|---------|
> | **Direct_GPT-4o** | 72.0 | 68.8 | 80.8 | 79.2 | 91.4 | 96.0 | 81.4 |
> | **InContext_GPT-4o** | 83.8 | 74.4 | 96.8 | 83.8 | 89.0 | 99.2 | 87.7 |
> | **SimVLM** | 93.7 | 94.7 | 83.2 | 86.0 | 84.6 | 71.2 | 85.6 |

---

> > ### Author Response · Authors · 2025-11-23
> > **Response to Reviewer 8x3c (Q3, Q4)**
> >
> > **8x3c-Q3: In Table 2, the CodePDDL baseline shows comparable performance to VLMFP on the Frozenlake and Maze domains but performs poorly on others. What is the underlying reason for this significant performance disparity across domains?**
> >
> > 8x3c-A3:
> > The fundamental reason for this significant performance is that Frozenlake and Maze have **fewer actions, simpler mechanisms, and less object types** than other domains.
> >
> > We provide a complexity comparison between different domains here:
> > * Frozenlake:
> >     * 4 actions: move up, move down, move left, move right
> >     * Mechanism: no stepping into ice holes
> >     * Object types: grid positions
> > * Maze:
> >     * 4 actions: move up, move down, move left, move right
> >     * Mechanism: cannot move into walls
> >     * Object types: grid positions
> > * Sokoban:
> >     * 3 actions: move, push-to-goal, push-to-nongoal
> >     * Mechanism: movable objects; can only move/push to clear a position; the agent, box, and goal positions need to be three consecutive positions in one direction
> >     * Object types: grid positions, direction, box
> > * Printer:
> >     * 8 actions: turn-left, turn-right, move, pick, drop, drop_desk, toggle_on, toggle_off
> >     * Mechanism: orientation; interactive objects
> >     * Object types: grid positions, direction
> > * Package:
> >     * 5 actions: turn-left, turn-right, move, open, close
> >     * Mechanism: orientation; interactive objects; multiple objects to visit
> >     * Object types: grid positions, direction, package
> > * Overcooked:
> >     * 9 actions: move, pick-ingredient, drop-ingredient, pick-plate, drop-plate, chop, merge-ingredient, put-plate, deliver
> >     * Mechanism: multiple players; interactive objects; objects in different states; multiple objects to visit
> >     * Object types: grid positions, direction, player, ingredient, plate, chopping-board, delivery-point
> >
> > When the domains become complex, without feedback and updating, it is easy for LLMs to ignore some preconditions/effects of actions or fail to assign every object a type, making it hard to directly generate correct domain and problem PDDL files. This significant performance difference showcases the importance of the feedback and updating modules when handling complex domains. We have also included this analysis in Appendix B.3 and thank the reviewer for bringing this up.
> >
> > We would also like to emphasize that, as described in line 400-401, CodePDDL is the baseline which prompts LLM to generate PDDL files **given problem descriptions generated by SimVLM**. Thus, although CodePDDL has comparable performance to VLMFP for simple domains, it still relies on fine-tuned SimVLM. As shown in the previous answer 8x3c-A2, using GPT-4o directly as SimVLM has a poor performance of 3.3%. This also demonstrates the effectiveness of SimVLM and the importance of the dual-VLM framework.
> >
> > **8x3c-Q4: Several relevant papers mentioned in the text (Line 52-79) are not included as baselines in Table 1 or 2. Including these comparisons is crucial for a fair and thorough evaluation of VLMFP's performance against contemporary works.**
> >
> > 8x3c-A4:
> > Our framework, to our knowledge, is the first framework to leverage **visual inputs to generate both PDDL files without human feedback or direct environment access.**
> >
> > As described in lines 52-79 and lines 123-139, these relevant works either
> > * require textual scenario descriptions, environment access, or pre-defined PDDL files, which **cannot be directly achieved through visual inputs**, or
> > * use VLM to describe the scenario, **assuming access to ground truth domain PDDL files**
> >
> > These limitations and assumptions make these works **not comparable** with VLMFP (results in Table 2).
> >
> > We notice that the reviewer also mentions to compare for Table 1 (results of SimVLM). There are two works that enable visual inputs [1,2].
> >
> > Work [1] uses an Open-Vocabulary Object Detector to serve as object detectors, thus it is not capable of simulating actions or judging goal reaching, which are important metrics in Table 1 and critical to the feedback and updating processes of VLMFP.
> >
> > Work [2] uses GPT-4o to describe the object's initial states and goal states, including an example of the desired output format. This baseline is added as GPT4o-InContext in 8x3c-A2, which still lacks accurate scenario and spatial understanding capabilities.
> >
> > We refer to Appendix C.4 for a failure example of using GPT-5 (even stronger than GPT4o) to accomplish the task.
> >
> >
> > [1] Shirai, Keisuke, et al. "Vision-language interpreter for robot task planning." 2024 IEEE International Conference on Robotics and Automation (ICRA). IEEE, 2024.
> >
> > [2] Dang, Xuzhe, Lada Kudláčková, and Stefan Edelkamp. "Planning with vision-language models and a use case in robot-assisted teaching." arXiv preprint arXiv:2501.17665(2025).

---

### Official Review · Reviewer_TrTP · 2025-11-03

**Soundness:** 2
**Presentation:** 2
**Contribution:** 3
**Rating:** 4
**Confidence:** 3

**Summary:**

This paper proposes a dual-VLM framework that generates and iteratively refines PDDL from visual inputs using SimVLM as a world model; the idea is simple, technically sound, and the experiments are well-designed. The approach is well-motivated and potentially generalizable.
However, the  evaluation is limited to grid-worlds, leaving scalability to continuous/3D/POMDP settings unclear. Convergence is under-specified: EW score threshold, iteration limits, failure handling when the score doesn’t reach 1.0, and reproducibility under stochastic simulations. In adition, one risk is hallucination and safety—SimVLM-consistency is not correctness without an external oracle, so hallucinations can yield confidently wrong PDDL.

**Strengths:**

This work proposes a dual-VLM framework that can generate more accurate PDDLs given visual observations and raw language prompts. The idea is simple yet effective: leverage an additional VLM as a world model (SimVLM), which generates and iteratively refines PDDL files by comparing their execution with SimVLM's simulations.
Overall, the paper makes meaningful contributions to PDDL generation from visual inputs.
The dual-VLM architecture is well-motivated, addressing the complementary weaknesses of spatial reasoning and PDDL generation. This idea of dual-VLM self-refinement can potentially be extended to VLM-based planning frameworks to improve planning domain knowledge generation.
However, there are some concerns about (1) task generalizability, (2) convergence analysis, and (3) hallucination and safety issues.

**Weaknesses:**

### Limited Tasks
The evaluation is restricted to grid-world domains. Real-world planning often involves continuous spaces, 3D environments, and more complex dynamics. The paper doesn't discuss how the approach would scale to such scenarios. For example, SimVLM may have issues for cases involving POMDPs.
### Hallucination Issues Without Guarantees
The framework fundamentally assumes SimVLM provides accurate ground truth for action simulations. However, this framework does not provide a proof or verification step against an external oracle; it guarantees syntactic validity and SimVLM-consistency, not alignment with real dynamics. When SimVLM hallucinates, the entire framework can produce incorrect PDDL files that match the hallucinated behavior rather than actual dynamics. The paper provides no mechanism to detect or correct SimVLM hallucinations, creating a critical failure mode where the dual-VLM framework could perform worse than simpler baselines. This is particularly concerning given recent work showing VLMs' propensity for confident hallucinations in spatial reasoning tasks.
### EW Score and Iterations Unclear
The paper is unclear about the refinement process. Specifically: (1) The threshold for acceptable alignment is never explicitly defined (only implied to be 1.0 from figures), (2) There is no discussion of what happens when convergence fails after multiple iterations (e.g., when the EW score cannot achieve 1.0), (3) Since SimVLM uses ChatGPT-4o which may have randomness in simulation, how is reproducibility ensured? To clarify these details, it would be better to provide: explicit convergence criteria, iteration limits, failure handling procedures, and empirical analysis of convergence rates across different domains.

**Questions:**

**SimVLM Architecture Choices:** Why was Qwen2-VL-7B chosen specifically? Have you experimented with other VLM architectures? An ablation study comparing different base models would be valuable.

**Failure Analysis:** Can you provide more detailed analysis of failure modes? When VLMFP fails, is it typically due to SimVLM perception errors, GenVLM generation errors, or convergence issues?

**Model Hallucination Issue:** If SimVLM hallucinates and creates inaccurate transitions, could the dual-VLM framework potentially end up with worse generation results? Is there any way to guarantee correctness?

**EW Score and Iterations:** Please see the questions outlined in the weakness section above.

---

> ### Author Response · Authors · 2025-11-23
> **Response to Reviewer TrTP (Summary and Q1)**
>
> We thank the reviewer for the constructive comments and helpful feedback! You brought up some great questions and suggestions, which have helped improve our work.
>
> We
> * Added an additional 3D assembly task to show that the proposed **VLMFP framework is capable of solving complex task planning in 3D domains** (TrTP-A1)
> * Added SimVLM base model ablation to show that the performance of **SimVLM is stable across different base models** (TrTP-A4)
> * Evaluated SimVLM across different seeds to show that **SimVLM is empirically stable across various seeds, greatly outperforming the baselines** (TrTP-A2)
> * Added additional implementation details (TrTP-A3) and failure mode analysis (TrTP-A5)
>
> Please also see the **Revision Summary** for additional experiments suggested by other reviewers. We address the reviewer's concerns below. We also updated a revised draft and colored the modifications/additional results and discussions with blue.
>
> **TrTP-Q1: The evaluation is restricted to grid-world domains. Real-world planning often involves continuous spaces, 3D environments, and more complex dynamics. The paper doesn't discuss how the approach would scale to such scenarios. For example, SimVLM may have issues for cases involving POMDPs.**
>
> TrTP-A1:
> We thank the reviewer for the valuable suggestions. We claim and show with **experiments of a 3D assembly task** that **VLMFP is not restricted to grid-world domains**, and it has the potential to scale to 3D continuous tasks, even with partially observable environments.
> To prove our claim, we evaluate our framework with a **complex long-horizon 3D assembly task** (from Reflective Planning [1]). We updated our draft to include this experiment and kindly refer the reviewer to Section 5.6 and Appendix C.7 for **visualizations** and details of the task.
> * Domain description
>     * In the scenario, there is a puzzle consisting of a board and several blocks with different colors on the table. The goal is to assemble the puzzle to reach the goal configuration with the robot arm. The robot arm can pick up, put down, insert, or reorient the block. The blocks have dependence on other blocks; that is, to achieve the goal, the blocks should be inserted after the blocks they depend on are inserted.
> * Challenges compared to existing grid-world evaluations
>     * **3D continuous realistic task**
>     * **No natural language description for goal configuration.** Since the Reflective Planning paper uses a goal image, instead of a clear natural language description, to represent the goal configuration, which is a much more challenging task, as the VLM would need to recognize, summarize, or even infer the dependency relationships between blocks of goal configuration. We keep this challenging setup to show the flexibility of SimVLM and VLMFP. Thus, SimVLM takes in both initial and goal configuration images as the inputs.
>     * **All are randomized**: block number, color, positions, shape, state; dependence order;  board color, position, shape;
>     * **Occlusion, thus partially observable.** Although not always, sometimes the stacked blocks could occlude each other.
> * Evaluation and analysis
>     * SimVLM: we collected ~25k datapoints and fine-tuned Qwen2-VL-7B as our SimVLM. The appearance differences are created through randomizing color/ position/ order/ number with different seeds. Similar to our previous evaluation, we tested our SimVLM with 1000 datapoints each from seen (S) and unseen (U) appearances.
>     * As shown in the table, SimVLM achieves consistently strong performance across all four evaluation metrics, with an average of 96.0% and 84.8% for seen and unseen appearances. The only visible decline appears in task-description accuracy on unseen appearances, but this is largely an artifact of our strict exact-match evaluation. Thus, even if a single sentence of description is missing, the description would be considered a failure. However, omissions in high-level descriptions do not necessarily affect downstream PDDL generation, since many objects mentioned in the description may be irrelevant to the specific plan. This is reflected in VLMFP’s end-to-end performance: despite the strict scoring on description matching, the overall planning success remains high across both seen and unseen settings.
>     * VLMFP: Similarly to the paper, we evaluate success rates for 1500 trials each for S and U. The success rates are **92.0%** and **85.0%**, respectively.
>
> **Table 1: String matching rate (%) of SimVLM and Success rate (%) of VLMFP on Assembly.**
>
> | Domain   | Task Description | Execution Reason | Execution Result | Goal Reach | Average | VLMFP |
> |----------|---------------|--------------|--------------|------------|---------|-------|
> | Assembly(S)| 86.9          | 99.2         | 99.2         | 98.6       | 96.0    | 92.0  |
> | Assembly(U)| 52.9          | 94.9         | 95.4         | 96.0       | 84.8    | 85.0  |
>
> TrTP-A1 continue next page

---

> > ### Author Response · Authors · 2025-11-23
> > **Response to Reviewer TrTP (Q1 -- continue)**
> >
> > TrTP-A1 - continue:
> > * Are 2D gridworld scenarios simpler?
> >     * **For PDDL-style task planning, complexity does not grow primarily with spatial dimensionality (2D vs. 3D), but with the richness of the symbolic structure being modeled.** In our framework, complexity comes from
> >         * (1) **scenario understanding and action simulation**, which depend on factors such as the number of objects, object types, spatial relations, etc.; and
> >         * (2) **domain formalization**, which depends on the complexity of action mechanisms, object interactions, rule variations, etc.
> >
> >     These sources of complexity exist regardless of whether the scene is rendered in 2D or 3D. To show the flexibility of our framework, we focus on **diverse domains with heterogeneous mechanisms, action types, and rule structures**, which represents complexity that is ***orthogonal*** to geometric dimensionality.
> > * Potential future extension
> >     * We believe that extending our framework to handle more diverse and realistic 3D tasks is an important direction for future work, particularly given the rich availability of real-world 3D vision and robotics datasets. We demonstrate that SimVLM can process both the initial and goal images as inputs. This suggests that the model can naturally extend to **multi-view inputs for handling occlusions** and to **temporal image sequences** that capture how a scene evolves over time, enabling planning in dynamic environments. We would love to explore these directions further in the future. However, we would like to emphasize that **establishing a reliable and generalizable visual-to-symbolic planning pipeline is itself a significant step for VLM-based long-horizon planning**, and it lays a solid foundation upon which many future extensions can be built.
> >
> > [1] Feng, Yunhai, et al. "Reflective planning: Vision-language models for multi-stage long-horizon robotic manipulation." Proceedings of the 9th Conference on Robot Learning (CoRL), 2025.

---

> ### Author Response · Authors · 2025-11-23
> **Response to Reviewer TrTP (Q2)**
>
> **TrTP-Q2: If SimVLM hallucinates and creates inaccurate transitions, could the dual-VLM framework potentially end up with worse generation results? Is there any way to guarantee correctness?**
>
> TrTP-A2:
> We agree that, without an external oracle, absolute correctness cannot be guaranteed. However, we conduct experiments to evaluate SimVLM across **3 random seeds** to measure its stability. As shown in the table below, across the 6 domains, the standard deviation for all four metrics is very small, indicating that **SimVLM’s perception, step-wise reasoning, and action-result predictions are highly stable and SimVLM does not hallucinate unpredictably**. We have added this multi-seed experiment result to Appendix C.3.
>
> Additionally, we emphasize that the SimVLM metrics are computed over **1000 evaluation datapoints**, and VLMFP results are averaged over **1500 planning trials** for each domain. These large-sample evaluations already demonstrate strong empirical stability despite the possibility of hallucination raised by the reviewer, and the resulting performance substantially exceeds all baseline models.
>
> We also evaluated GPT4o (with in-context example) as the SimVLM and showed that it can only provide, on average, 3.3% accurate scenario descriptions across 6 domains, compared to 82.6% of our fine-tuned VLM. This shows that API-based large VLMs **lack precise spatial understanding and long-horizon reasoning capabilities** to directly accomplish the task of SimVLM. We kindly refer the reviewer to Appendix C.1 for more detailed tables across four tables for this added baseline.
>
> **Table 2: The mean of string matching rate (%) for 4 SimVLM output types on 6 grid-world domains across 3 seeds.**
>
> | Output           | Frozenlake S | Frozenlake U | Maze S | Maze U | Sokoban S | Sokoban U | Package S | Package U | Printer S | Printer U | Overcooked S | Overcooked U | Avg S | Avg U |
> |------------------|--------------|--------------|--------|--------|-----------|-----------|-----------|-----------|-----------|-----------|--------------|--------------|--------|--------|
> | Task Descrip.    | 99.9         | 91.9         | 99.2   | 89.9   | 74.0      | 51.4      | 100.0     | 99.9      | 99.8      | 96.6      | 99.2         | 64.9         | **95.3**   | **82.4**   |
> | Exec. Reason     | 94.9         | 94.8         | 83.6   | 96.9   | 72.4      | 83.4      | 88.5      | 89.0      | 85.7      | 85.4      | 76.9         | 75.8         | **83.7**   | **87.5**   |
> | Exec. Result     | 94.9         | 94.7         | 83.6   | 97.0   | 72.1      | 83.0      | 88.5      | 89.0      | 85.7      | 85.4      | 77.0         | 75.9         | **83.7**   | **87.5**   |
> | Goal Reach       | 93.5         | 93.5         | 80.3   | 94.7   | 72.7      | 83.1      | 86.8      | 86.6      | 83.6      | 83.7      | 73.9         | 73.2         | **81.8**   | **85.8**   |
> | **Average**      | **95.8**     | **93.7**     | **86.7** | **94.6** | **72.8** | **75.2** | **91.0** | **91.1** | **88.7** | **87.8** | **81.8**     | **72.4**     | **86.1** | **85.8** |
>
> **Table 3: The standard deviation of string matching rate (%) for 4 SimVLM output types on 6 grid-world domains across 3 seeds.**
>
> | Output           | Frozenlake S | Frozenlake U | Maze S | Maze U | Sokoban S | Sokoban U | Package S | Package U | Printer S | Printer U | Overcooked S | Overcooked U | Avg S | Avg U |
> |------------------|--------------|--------------|--------|--------|-----------|-----------|-----------|-----------|-----------|-----------|--------------|--------------|--------|--------|
> | Task Descrip.    | 0.1          | 0.5          | 0.1    | 0.1    | 2.1       | 0.3       | 0.0       | 0.2       | 0.1       | 0.4       | 0.2          | 1.1          | **0.3**    | **0.2**    |
> | Exec. Reason     | 1.5          | 1.3          | 1.5    | 0.2    | 3.8       | 2.0       | 2.3       | 1.0       | 2.2       | 2.0       | 1.8          | 3.7          | **2.0**    | **1.3**    |
> | Exec. Result     | 1.4          | 1.3          | 1.5    | 0.2    | 3.6       | 0.6       | 2.3       | 1.0       | 2.2       | 2.0       | 1.6          | 3.7          | **2.0**    | **1.0**    |
> | Goal Reach       | 0.5          | 0.2          | 1.7    | 0.0    | 1.4       | 0.1       | 1.1       | 0.7       | 0.5       | 0.8       | 1.9          | 2.4          | **0.7**    | **0.2**    |
> | **Average**      | **0.8**      | **0.7**      | **0.7** | **0.0** | **2.2**   | **0.6**   | **1.4**   | **0.7**   | **1.2**   | **1.1**   | **0.9**      | **2.5**      | **1.1** | **0.6** |

---

> ### Author Response · Authors · 2025-11-23
> **Response to Reviewer TrTP (Q3)**
>
> **TrTP-Q3: EW Score and Iterations details**
>
> TrTP-A3: We thank the reviewer for pointing out some missing details. We provide the details here and also have included this information in Appendix D.3 in our draft.
> The updating loop is skipped when the EW score reaches 1.0(`abs(ew_rating - 1.0) < 1e-6`) and the plan generated using current PDDL files is valid. This means, the generated PDDL files are returned when they are verified by SimVLM to be able to solve the current instance and can achieve the same results as SimVLM for all explorations.
> The iteration repeats for 4 times, and if the EW score cannot achieve 1.0 or the generated plan is not valid, the generated PDDL files are returned. However, this does not necessarily imply that the generated files are entirely incorrect. For example, when the problem PDDL file is incorrect but the domain PDDL is correct, a well-specified domain PDDL generalizes across instances, enabling potential successful solutions for other instances.
> We would like to clarify that the SimVLM does not use GPT-4o. Instead, it is our fine-tuned 7B model on fixed-format examples. The experiment in the previous section also empirically shows the stability of SimVLM across different domains and seeds.
> We collected the convergence rates for all 6 domains. Each is evaluated on 15 problem instances, with a maximum round of 4. On average, 77.1% and 64.8% problem instances are converged for seen and unseen appearances, respectively. These results show that the EW-guided refinement loop is effective across diverse domains and appearance variations. We find that convergence tends to be fastest in domains with simpler transition rules (e.g., FrozenLake, Maze) and more gradual in domains with richer multi-object interactions (e.g., Sokoban, Overcooked), which is consistent with our failure-mode analysis in TrTP-A5.
>
> **Table 4: Convergence Rate (\%) of VLMFP for 15 input instances on 6 domains.**
>
> | Convergence   | Frozenlake | Maze | Sokoban | Package | Printer | Overcooked | Average |
> |----------|---------------|--------------|--------------|------------|------------|---------|-------|
> | VLMFP(S)| 100.0          | 93.3         | 66.7         | 73.3       | 66.7    | 40.0  |77.1  |
> | VLMFP(U)| 86.7          | 80.0         | 33.3         | 53.3       | 66.7    | 33.3  | 64.8  |

---

> ### Author Response · Authors · 2025-11-23
> **Response to Reviewer TrTP (Q4)**
>
> **TrTP-Q4: Why was Qwen2-VL-7B chosen specifically? Have you experimented with other VLM architectures? An ablation study comparing different base models would be valuable.**
>
> TrTP-A4:
> We selected Qwen2-VL-7B as it is one of the small VLMs that achieves state-of-the-art performance across a wide spectrum of multimodal benchmarks. As suggested by the reviewer, we added an ablation study that evaluates SimVLM performance, comparing Qwen2-VL-7B with LLaVA-NeXT-7B and Google PaliGemma2-10B. The results are shown in the following tables. Both models performed reasonably well, but they showed slightly lower accuracy, higher variance, and were less generalizable to unseen appearances than Qwen2-VL-7B. Importantly, although the choice of Qwen2-VL-7B reflects empirical stability, our framework does not depend on Qwen2-VL-7B specifically. VLMFP remains model-agnostic, and substituting the simulator backbone is straightforward. We added this ablation experiment in Appendix C.2.
>
> **Table 5: String matching rate (%) for 4 SimVLM output types on 6 grid-world domains, with LLaVA-NeXT-7B as the base model.**
>
> | Output           | Frozenlake S | Frozenlake U | Maze S | Maze U | Sokoban S | Sokoban U | Package S | Package U | Printer S | Printer U | Overcooked S | Overcooked U | Avg S | Avg U |
> |------------------|--------------|--------------|--------|--------|-----------|-----------|-----------|-----------|-----------|-----------|--------------|--------------|--------|--------|
> | Task Descrip.    | 99.3         | 87.2         | 99.5   | 30.7   | 85.9      | 61.2      | 99.0      | 98.6      | 99.7      | 80.7      | 97.0         | 43.0         | **96.7**   | **66.9**   |
> | Exec. Reason     | 89.2         | 86.2         | 74.5   | 91.2   | 64.2      | 81.2      | 79.5      | 79.2      | 75.3      | 75.7      | 68.5         | 68.5         | **75.2**   | **80.3**   |
> | Exec. Result     | 89.3         | 86.4         | 74.2   | 91.3   | 64.9      | 82.3      | 79.6      | 79.2      | 75.4      | 75.8      | 68.8         | 68.8         | **75.4**   | **80.6**   |
> | Goal Reach       | 89.6         | 87.6         | 73.9   | 91.2   | 65.9      | 83.5      | 80.1      | 79.7      | 75.9      | 76.3      | 69.2         | 69.2         | **75.8**   | **81.2**   |
> | **Average**      | **91.8**     | **86.8**     | **80.5** | **76.1** | **70.2** | **77.0** | **84.5** | **84.2** | **81.6** | **77.1** | **75.9**     | **62.4**     | **80.8** | **77.3** |
>
> **Table 6: String matching rate (%) for 4 SimVLM output types on 6 grid-world domains, with PaliGemma2-10B as the base model.**
>
> | Output           | Frozenlake S | Frozenlake U | Maze S | Maze U | Sokoban S | Sokoban U | Package S | Package U | Printer S | Printer U | Overcooked S | Overcooked U | Avg S | Avg U |
> |------------------|--------------|--------------|--------|--------|-----------|-----------|-----------|-----------|-----------|-----------|--------------|--------------|--------|--------|
> | Task Descrip.    | 99.8         | 88.5         | 99.6   | 13.2   | 76.7      | 50.6      | 95.2      | 94.4      | 88.1      | 85.8      | 99.7         | 42.5         | **93.2**   | **62.5**   |
> | Exec. Reason     | 90.9         | 87.2         | 77.5   | 94.0   | 68.5      | 84.7      | 83.0      | 82.8      | 78.9      | 78.8      | 72.5         | 72.8         | **78.6**   | **83.4**   |
> | Exec. Result     | 90.9         | 87.2         | 77.6   | 94.1   | 67.3      | 86.4      | 83.2      | 83.0      | 79.0      | 78.9      | 73.3         | 73.5         | **78.6**   | **83.9**   |
> | Goal Reach       | 90.9         | 89.7         | 77.2   | 94.0   | 71.0      | 87.2      | 83.1      | 82.8      | 80.0      | 79.7      | 74.1         | 74.1         | **79.4**   | **84.6**   |
> | **Average**      | **93.2**     | **88.1**     | **83.0** | **73.9** | **70.9** | **77.2** | **86.1** | **85.8** | **81.5** | **80.8** | **79.9**     | **65.7**     | **82.4** | **78.6** |

---

> > ### Author Response · Authors · 2025-11-23
> > **Response to Reviewer TrTP (Q5)**
> >
> > **TrTP-Q5: Can you provide more detailed analysis of failure modes? When VLMFP fails, is it typically due to SimVLM perception errors, GenVLM generation errors, or convergence issues?**
> >
> > TrTP-A5:
> > We would first clarify that the convergence issue mentioned by the reviewer is either due to SimVLM perception errors or GenVLM generation errors. Thus, these two errors are the major failure modes. We looked into the failure reasons of the 15 input instances for both seen and unseen appearances for all domains and summarized them below. Values show (errors / total failures) for each category.
> >
> > **Table 7: Error type distribution of VLMFP for 15 input instances on 6 domains.
> > Values show errors / total_failures for each category.**
> >
> > | Error Type | Frozenlake S | Frozenlake U | Maze S | Maze U | Sokoban S | Sokoban U | Package S | Package U | Printer S | Printer U | Overcooked S | Overcooked U |
> > |------------|--------------|--------------|--------|--------|-----------|-----------|-----------|-----------|-----------|-----------|--------------|--------------|
> > | Perception | 0/0          | 0/1          | 0/1    | 1/3    | 2/5       | 5/8       | 0/4       | 0/7       | 0/5       | 0/5       | 0/8          | 5/10         |
> > | Generation | 0/0          | 1/1          | 1/1    | 2/3    | 3/5       | 3/8       | 4/4       | 7/7       | 5/5       | 5/5       | 8/8          | 5/10         |
> >
> >
> > From the table, we can observe that for **visually busier scenes**(Sokoban) and scenarios with **unseen objects** (onions in Overcooked U), the perception error becomes more common. While in other cases where the **domain mechanisms are complex with various constraints**, failing to generate and update correct action specifications, necessary predicates, or initializing objects within limited rounds is the major failure case.
> >
> > We believe that more techniques could be implemented to mitigate both perception errors and generation errors. We observe during the development of VLMFP that the generalization capability of SimVLM improved as it sees more diverse domains, map sizes, appearances, and rules. Thus, there is great potential to develop a more generalized simulation ‘world model’ when it exposes stronger VLM base models to increasingly diverse environments. Additionally, the errors of GenVLM could decrease with more PDDL-targeted prompting and checkers, more updating rounds, and stronger VLM reasoning capabilities.
> >
> > There remain many promising directions, both engineering-oriented and research-oriented, that could further strengthen the framework. We leave these explorations to future work and emphasize that VLMFP is, to the best of our knowledge, the first framework to generate both PDDL domain and problem files directly from visual inputs, without human feedback or environment access. We believe that establishing a reliable and generalizable visual-to-symbolic planning pipeline is a substantial contribution to VLM-based long-horizon planning and opens numerous avenues for future research. We have added this discussion in Appendix D.2.

---

### Author Response · Authors · 2025-11-23
**Revision Summary: additional experiments, discussions, and draft revisions**

We thank all reviewers for their thoughtful comments and suggestions! To help reviewers to better access the updates we have made, we include this summary of revisions as below:

To summarize the additional experiments and discussions we added:
* Added an additional 3D assembly task to show that the proposed **VLMFP framework is capable of solving complex task planning in 3D domains.**
    * SimVLM achieves consistently strong performance across all four evaluation metrics, with an average of **96.0%** and **84.8%** for seen and unseen appearances.
    * VLMFP achieves success rates of **92.0%** and **85.0%**, respectively, evaluated on 1500 trials each for seen and unseen appearances.
    * Section 5.6, Appendix B.1, Appendix C.7
* Conducted baseline comparisons for SimVLM
    * API-based large VLMs **lack precise spatial understanding and long-horizon reasoning capabilities**
    * Appendix C.1
* Added SimVLM base model ablation to show that the performance of **SimVLM is stable across different base models**
    * SimVLM with other base models performed reasonably well, with slightly worse performance than Qwen2-VL-7B
    * VLMFP remains model-agnostic. Substituting the simulator backbone is straightforward.
    * Appendix C.2
* Evaluated SimVLM across different seeds to show that **SimVLM is empirically stable across various seeds, greatly outperforming the baselines**
    * Standard deviation for all four metrics is very small, with an average of 1.1% and 0.6% for seen and unseen appearances.
    * Appendix C.3
* Added domain complexity analysis, additional implementation details, runtime analysis, and failure mode analysis
    * Appendix B.3, D.2, D.3, D.4

In addition to the above additions, we revised the following parts of the main paper to make the presentation clearer:
* We added a high-level conceptual overview of VLMFP to provide an abstract, conceptual, and clean narrative of VLMFP and its objective. (Section 3.2)
* We modified the narratives within subsections of the method section to make it more logical and smooth.  (Section 3.1, Section 3.3)
* We separated the implementation details to be another section. (Section 4)
* We add more citations.  (Section 2)

---

### Meta-Review · Area_Chair_1uCz · 2025-12-26

**Summary:**

This paper introduces a framework for translating a visually defined agent environments (2D grid worlds and a simple 3D environment) into a Planning Domain Definition Language (PDDL). The framework uses two separate VLMs, one VLM (SimVLM) to simulate action consequences given rule descriptions, and another VLM to generate and refine the PDDL. Concerns were raised about clarity of the paper and about simplicity of the 2D grid world environments, but both were successfully addressed in the author rebuttal. Experimental results confirm the viability of the approach.

Overall, the AC believes that the paper can be accepted given that the authors addressed the main reviewer concerns and overall improved clarity of the paper. The AC recommends carefully assessing claims about the necessity of splitting the model into multiple different VLMs (highlighted by the reject voting reviewer). It would probably make sense to frame this as a convenient, empirically motivated choice for this particular project but not as a broader necessity.

**Reviewer Concerns:**

Two weak reject voting reviewers: primary concern is limited application domain (2D grid worlds). This was successfully addressed during the rebuttal: the authors added convincing results on a complex 3D environment. The AC expects that these two reviewers would have revised their score to weak accept or higher accordingly.

The reject voting reviewer primarily highlights a lack of writing/exposition clarity as reason for rejection, while otherwise finding the paper of interest to the community, sufficiently novel, and highlights the positive results.

The AC read the paper and finds the writing in the revised version to be sufficiently clear. The authors do, however, make some questionable general statements in their rebuttal like “splitting into two VLMs is necessary as they have different and conflicting advantages, and the small VLM and large VLM cannot replace each other”. This might be true for the specific model variants/checkpoints used in this paper, but the statement could be understood as a broad general statement — which seems unsubstantiated and goes against the general understanding of this class of models in the ML/AI community (which in turn would require much stronger substantiation).

In summary, the reject voting reviewer would have likely still voted for at least a weak reject and not raised the score into “accept” territory.

**Reviewer Scores:**

See above: no change in the accept/reject voting reviewers, but likely an improvement in score in the two weak reject voting reviews, placing them in (weak) accept territory.

---

### Decision · Program_Chairs · 2026-01-26

Accept (Poster)